# Data Distribution Valuation

**Xinyi Xu**
Department of Computer Science
National University of Singapore
`xinyi.xu@u.nus.edu`

**Shuaiqi Wang**
Department of Electrical and Computer Engineering
Carnegie Mellon University
`shuaiqiw@andrew.cmu.edu`

**Chuan-Sheng Foo**
Institute for Infocomm Research
Agency for Science, Technology and Research
`foo_chuan_sheng@i2r.a-star.edu.sg`

**Bryan Kian Hsiang Low**
Department of Computer Science
National University of Singapore
`lowkh@comp.nus.edu.sg`

**Giulia Fanti**
Department of Electrical and Computer Engineering
Carnegie Mellon University
`gfanti@andrew.cmu.edu`

## Abstract

Data valuation is a class of techniques for quantitatively assessing the value of data for applications like pricing in data marketplaces. Existing data valuation methods define a value for a discrete dataset. However, in many use cases, users are interested in not only the value of the dataset, but that of the *distribution* from which the dataset was sampled. For example, consider a buyer trying to evaluate whether to purchase data from different vendors. The buyer may observe (and compare) only a small preview sample from each vendor, to decide which vendor's data distribution is most useful to the buyer and purchase. The core question is *how should we compare the values of data distributions from their samples?* Under a Huber characterization of the data heterogeneity across vendors, we propose a maximum mean discrepancy (MMD)-based valuation method which enables theoretically principled and actionable policies for comparing data distributions from samples. We empirically demonstrate that our method is sample-efficient and effective in identifying valuable data distributions against several existing baselines, on multiple real-world datasets (e.g., network intrusion detection, credit card fraud detection) and downstream applications (classification, regression).

## 1 Introduction

*Data valuation* is a widely-studied practice of quantifying the value of data [57]. Today, data valuation methods define a value for a discrete dataset $D$, i.e., a fixed set of samples [24, 32]. However, many emerging use cases require a data user to evaluate the quality of not just a dataset, but the *distribution* from which the data was sampled. For example, data vendors in markets like Datarade and Snowflake, financial data streams [46, 52], information security data [22]) offer a preview in the form of a sample dataset to prospective buyers [4, 11]. Similarly, enterprises selling access to generative models may offer a limited preview of the output distributions to prospective buyers [53]. Buyers use these sample datasets to decide whether to pay for a full dataset or data stream—i.e., access to the data distribution. Concretely, the buyers would compare between different data distributions (via their respective sample datasets) to determine and select the more valuable one.

38th Conference on Neural Information Processing Systems (NeurIPS 2024).

In such applications, existing dataset valuation metrics are missing two components: (1) They do not formalize the value of the underlying sampling distribution, (2) nor do they provide a theoretically principled and actionable policy for comparing different sampling distributions based on the sample datasets. For example, most existing data valuation techniques are designed only to value a dataset [57]. To our knowledge, there are no methods designed to value an underlying distribution.

To model this problem, we consider a data buyer who wishes to evaluate $n$ data vendors, each with their own dataset $D_i$ drawn i.i.d. from distribution $P_i$, where $i \in [n]$. The vendors' distributions are *heterogeneous*, i.e., the $P_i$'s can differ across vendors. Such distributional heterogeneity can arise from natural variations in data [12] or adversarial data corruption [27, 65]. The buyer's goal is to select the vendor whose data distribution is closest in some sense (to be defined) to a reference distribution $P^*$, which is fixed but unknown. Our goal is to provide both a precise definition for the value of the vendor distribution $P_i$ with respect to $P^*$, as well as a corresponding valuation for the sample dataset $D_i \sim P_i$. In particular, we want precise conditions under which, given two datasets $D_i$ and $D_j$ drawn from distributions $P_i$ and $P_j$, respectively, we can conclude that $P_i > P_j + \varepsilon_\Upsilon$ for some user-specified $\varepsilon_\Upsilon > 0$ with fixed probability. While this problem is straightforward when different vendors have the same underlying distribution, the main challenge is accounting for the heterogeneity in the data.

We thus identify three technical and modeling challenges: *(i)* What is a suitable heterogeneity model that captures realistic data patterns, while also admitting theoretical analysis? *(ii)* How should one define the value of a distribution under a given heterogeneity model? *(iii)* Many existing data valuation methods use a reference $P^*$, either explicitly [24, 33, 55] or implicitly [1, 14]. However, there are practical difficulties: (1) different data vendors may disagree on the choice of reference [70], (2) such a reference may not be available *a priori* [14, 56], or (3) dishonest vendors may try to overfit to the reference [2]. To address this, some works consider alternatives (to $P^*$) based on the vendors' distributions $P_i$'s such as their union [60] or a certain combination [66], but without theoretical analysis or justifications for their specific choices. So what is a practical alternative, and how can we theoretically justify it?

To address these challenges, we make three key choices.

*(a) Heterogeneity model.* We assume that each vendor's data distribution $P_i$ is a Huber model [30], which is a mixture model of the unknown distribution $P^*$ and an arbitrary outlier distribution $Q_i$. While the Huber model does not capture all kinds of statistical heterogeneity, mixture models both are a reasonable model for data heterogeneity in practice [9, 54] and have deeper roots in robust statistics [20, 40]. More importantly, the Huber model enables a direct and precise characterization of the effect of heterogeneity on the value of data (under our design choices *(b)* and *(c)* below), which has not been considered in prior works [1, 14, 66].

*(b) Value of a sampling distribution.* We use the negative maximum mean discrepancy (MMD) [26] between a reference distribution $P^*$ and $P$ as the value of the sampling distribution $P$. Then, we leverage a (uniformly converging) MMD estimator [26] to derive actionable policies for comparing sampling distributions with theoretical guarantees. In other words, a buyer can compare (the values of) sampling distributions $P_i, P_{i'}$ (from vendors $i, i'$) based on the respective samples $D_i, D_{i'}$ to determine which is more valuable, and by how much.

*(c) Choice of reference data distribution and dataset.* Unlike in prior works (e.g., [24, 67]), we do not specify a reference distribution outright. Instead, we consider a class of convex mixtures of vendor distributions as the reference. We first derive error guarantees and an actionable comparison policy for using general convex mixture of the vendors' distributions as the reference. We then propose to use the special case of a uniform mixture, justified by a game-theoretic argument, stating that the uniform strategy is worst-case optimal in a two-player zero-sum game.

These design choices are not isolated answers to each technical challenge, but collectively address these challenges (e.g., the analytic properties of both Huber and MMD are needed to compare distributions effectively). Our specific contributions are summarized as follows:

- We formulate the problem of data distribution valuation in data markets and study an MMD-based method for data distribution valuation. Under a Huber model of data heterogeneity, we show that this data valuation metric admits actionable, theoretically-grounded policies for comparing sampling distributions from samples.

- We derive an error guarantee (Proposition 2) and comparison policy (Theorem 1) for a general convex mixture $P_\omega$ as a reference (in place of $P^*$), thus relaxing the common assumption of knowing $P^*$. We then identify the uniform mixture as a game-theoretic special case for $P_\omega$.
- We demonstrate on real-world classification (e.g., network intrusion detection) and regression (e.g., income prediction) tasks that our method is sample-efficient, and effective in identifying the most valuable sampling distributions against existing baselines. For example, on classification tasks, we observed that an MMD-based valuation method outperformed four leading valuation metrics on 3 out of 4 classification settings (Table 2).

## 2 Related Work

Existing dataset valuation methods fall roughly in 2 categories: those that assume a given reference dataset, and those that do not. We defer additional discussion to App. B due to space constraints.

**With a given reference.** Several existing methods require a *given reference* in the form of a validation set (e.g., [24, 39]) or a baseline dataset [2]. Data Shapley [24], Beta Shapley [39] and Data Banzhaf [43, 63] utilize the validation accuracy of a trained model as the value of the training data. Class-wise Shapley [55] evaluates the effects of a dataset on the in-class and out-of-class validation accuracies. Both LAVA [33] and DAVINZ [67] use a proxy for the validation performance of the training data as their value, instead of the actual validation performance, to be independent of the choice of downstream task ML model [33] or to remove the need for model training [67]. Differently, [2] assume that the buyer provides a baseline dataset as the reference to calculate a relevance score used to evaluate the vendor's data. Therefore, these methods *cannot* be applied without such a given reference, which can be difficult to obtain in practice [14, 56]. In contrast, our method can be applied without a given reference, by carefully constructing a reference (Sec. 4.2).

**Without a given reference.** To relax the assumption of a given reference, [14, 60, 66] construct a reference from the data from all vendors. While the settings of [14, 60, 66] can include heterogeneity in the vendors' data, they do not explicitly formalize it and thus cannot precisely analyze its effects on data valuation. In contrast, our method, via the careful design choices of the Huber model (for heterogeneity) and MMD (for valuation), uniquely offers a precise analysis on the effect of heterogeneity on the value of data (Eq. (2)). Furthermore, these methods did not provide theoretical guarantees on the error arising from using their constructed reference in place of the ground truth (i.e., $P^*$). In contrast, by exploiting Observation 1 in the setting of multiple vendors and the MMD-based valuation (Eq. (1)), we provide such theoretical guarantees (e.g., Proposition 2). In a different approach to relax the assumption of a given reference, [56, 70] remove the dependence on a reference; as a result they can produce counter-intuitive data values under heterogeneous data (experiments in Sec. 5). The closest related work to ours is [60], which adopts the $\text{MMD}^2$ as a valuation metric, primarily for computational reasons. However, this work does not consider data *distribution* valuation, nor does it describe how to compare (the values of) distributions, let alone with theoretical guarantees. This comparison (with $\text{MMD}^2$) is expanded in App. B.2.

Table 1 provides a comparison between our choice (MMD) and three others: the Kullback-Leibler (KL) divergence [1, 66], the Wasserstein distance (WD) [33], and squared MMD ($\text{MMD}^2$) [60].

## 3 Model, Problem Statement and MMD

We consider a set of data vendors $i \in N := \{1, \ldots, n\}$, each with a *sample* dataset $D_i := \{z_{i,1}, \ldots, z_{i,m_i}\}$ of size $m_i$, where $z_{i,j}$ is sampled i.i.d. from the distribution $P_i$ [10]. Slightly abusing notations, we write $D_i \sim P_i$. We assume the existence of an unknown ground truth distribution $P^*$, called the test distribution [24, 32], true data distribution [1] or the task distribution [2].

**Huber model.** We assume that each sampling distribution $P_i$ follows a Huber model [30], defined as follows: $P_i = (1 - \varepsilon_i)P^* + \varepsilon_i Q_i$ where $\varepsilon_i \in [0, 1)$ and $Q_i$ is a distribution that captures the heterogeneity of vendor $i$ [27]. For notational simplicity, we omit the subscript $i$ and write $D, P$ instead of $D_i, P_i$, when it is clear. We adopt the Huber model because (i) it is sufficiently general to model various sources of heterogeneity [12, 13]; (ii) Huber models are "closed" under mixtures (i.e., a mixture of Hubers is a Huber model), so we can define a mixture over data vendors' distributions:

**Observation 1.** For a mixture weight $\omega \in \triangle(n-1)$,[1] define the mixture $P_\omega := \sum_{i \in N} \omega_i P_i$. Then, $P_\omega = (1 - \varepsilon_\omega)P^* + \varepsilon_\omega Q_\omega$ where $\varepsilon_\omega = \sum_{i \in N} \omega_i \varepsilon_i$ and $Q_\omega = (1/\varepsilon_\omega) \sum_{i \in N} (\omega_i \varepsilon_i Q_i)$ .

The mixture distribution $P_\omega$ (of individual $P_i$'s) is a Huber model, which is used in the theoretical results in Sec. 4.2. Define $D_\omega$ as the sample dataset by randomly sampling from each $D_i$ w.p. $\omega_i$, so effectively $D_\omega \sim P_\omega$. In particular, if $\omega$ is uniform (i.e., $\forall i, \omega_i = 1/n$), we denote the corresponding $P_\omega$ as $P_U$, $D_\omega$ as $D_U$, $\varepsilon_\omega$ as $\varepsilon_U$ and $Q_\omega$ as $Q_U$. We further expand our considerations of the Huber model to characterize data heterogeneity w.r.t. existing works in App. B.3. Later, we also empirically investigate non-Huber settings where our method (in Sec. 4) remains effective (App. D.3.4).

**Problem statement.** Given two datasets $D \sim P$ and $D' \sim P'$, we seek a distribution valuation function $\Upsilon(\cdot)$ and a dataset valuation function $\nu(\cdot)$ which enable a set of conditions under which to conclude that $\Upsilon(P) > \Upsilon(P')$, given only $\nu(D)$ and $\nu(D')$. Moreover, we seek a practical implementation of $\Upsilon$ that does *not* require access to the ground truth distribution $P^*$ as reference or any prior knowledge about the vendors (except each $P_i$ is Huber).

Existing methods cannot be easily applied to solve this problem. First, existing methods (e.g., [24, 32] do not define $\Upsilon$; hence, they cannot analyze the conditions under which $\Upsilon(P) > \Upsilon(P')$. In App. C, we elaborate why a dataset valuation $\nu$ cannot be easily extended to a data distribution valuation $\Upsilon$ and also highlight the theoretical appeal of directly considering $\Upsilon$ instead. Additionally, methods that explicitly require access to the reference distribution via $D^* \sim P^*$ (e.g., [39, 55, 67]) *cannot* be applied here. For other methods, additional non-trivial assumptions (e.g., [14, Assumption 3.2], [66, Assumption 3.1]) are required; we elaborate on these in App. B.1.

### 3.1 Maximum Mean Discrepancy (MMD)

The MMD is an integral probability metric proposed to test if two distributions are the same.

**Definition 1** (MMD, [26, Definition 2]). For a class $\mathcal{F}$ of functions $f$ in the unit ball of the reproducing kernel Hilbert space associated with a kernel function $k$, the MMD, which is symmetric, between two distributions $P, P'$ is $d(P, P'; \mathcal{F}) := \sup_{f \in \mathcal{F}} \mathbb{E}_{X \sim P} f(X) - \mathbb{E}_{W \sim P'} f(W)$ .

The MMD has a (biased) estimator for $D \sim P$ and $D' \sim P'$, and $|D| = m, |D'| = m'$ [26, Eq. (5)]: $\hat{d}(D, D') := [\frac{1}{m^2} \sum_{x,x' \in D} k(x, x') - \frac{2}{mm'} \sum_{(x,w) \in D \times D'} k(x, w) + \frac{1}{m'^2} \sum_{w,w' \in D'} k(w, w')]^{0.5}$ . Importantly, this estimator satisfies uniform convergence (Lemma 1), which is used in our theoretical results (e.g., Proposition 1). We denote with $K$ an upper bound on the kernel $k$: $\forall x, x', k(x, x') \leq K$. As $\mathcal{F}$ is associated with $k$ and kept constant throughout, its notational dependence is suppressed.

## 4 MMD-based Data Distribution Valuation

A distribution valuation function should intuitively reward distributions $P$ that are "closer" to the reference distribution $P^*$; accordingly, it should assign greater reward to datasets $D \sim P$ that are drawn from distributions that are closer to $P^*$. We study the following data distribution valuation:

$$\Upsilon(P) := -d(P, P^*) , \quad \nu(D) := -\hat{d}(D, D^*) . \tag{1}$$

To interpret, the value $\Upsilon(P)$ of a vendor's sampling distribution $P$ is defined as the negated MMD between $P$ and $P^*$, while the value $\nu(D)$ of its sample dataset $D$ is defined as the negated MMD estimate between $D$ and the reference dataset $D^*$.

**On the choice of MMD.** We summarize a comparison with three alternatives (i.e., KL-divergence (KL), Wasserstein Distance (WD), and MMD$^2$) in Table 1 to highlight the suitability of MMD for data distribution valuation. Despite its wide adoption, KL is difficult to estimate, and the available estimator only has asymptotic convergence guarantees [64] (rather than a finite-sample result), which can be arbitrarily slow. Its implementation also suffers from the curse of dimensionality [59, 64]. In addition, KL does not satisfy the triangle inequality, which our proof technique uses in Proposition 2. WD, also known as the optimal transport (OT) distance [34], suffers from the curse of dimensionality, as seen in the complexity results [23] in Table 1, and is more computationally costly to evaluate than MMD. MMD$^2$, though shares similar complexity results to MMD, does not satisfy desirable analytic properties, such as the triangle inequality or the property with Huber model. This comparison over divergences is expanded in App. B.2.

---

[1] $\omega$ is an $n$-dimensional probability vector in the $n-1$ simplex.

Table 1: Comparison with KL, WD and MMD$^2$, on sample and computational complexities, triangle inequality and connection with the Huber model. dim is the dimension of the random variable/data.

| | sample | computational | triangle inequality | Huber |
|---|---|---|---|---|
| KL | asymptotic | N.A. | ✗ | ✗ |
| WD | $\mathcal{O}(1/m^{1/\dim})$ | $\mathcal{O}(m^3 \log m)$ | ✓ | ✗ |
| MMD$^2$ | $\mathcal{O}(1/\sqrt{m})$ | $\mathcal{O}(m^2)$ | ✗ | ✗ |
| MMD | $\mathcal{O}(1/\sqrt{m})$ | $\mathcal{O}(m^2)$ | ✓ | ✓ |

MMD is both practically and theoretically appealing for data distribution valuation. Practically, MMD has lower sample and computational complexities in the dimension of the data, which is important because the real-world datasets can be complex and have a high dimension. Specifically, we leverage the uniform convergence (with sample complexity as in Table 1) of an MMD estimator to derive an actionable policy for comparing two distributions (i.e., Proposition 1, Theorem 1). Theoretically, MMD satisfies the triangle inequality, making it amenable to theoretical analysis, such as the derivation of our error guarantee (i.e., Proposition 2). Moreover, MMD pairs well with the Huber model in providing a precise characterization of the effect of heterogeneity on the value of data. In contrast, existing valuation works have *not* established a formal analysis on the heterogeneity (w.r.t. a specific choice for heterogeneity) on the value of data, elaborated in App. B.3.

**Effect of heterogeneity on data valuation.** Intuitively, the quality of a Huber distribution $P$ depends on both the size of the outlier component $\varepsilon$ and the statistical difference $d(P^*, Q)$ between $Q$ and $P^*$. A larger $\varepsilon$ and/or a larger $d(P^*, Q)$ decreases the value $\Upsilon(P)$. Our choice of MMD makes this intuition precise and interpretable: By Lemma 2, for $P = (1 - \varepsilon)P^* + \varepsilon Q$,

$$\Upsilon(P) = -\varepsilon d(P^*, Q) . \tag{2}$$

Eq. (2) shows that for a fixed $\varepsilon$, $P$'s value decreases linearly w.r.t. $d(P^*, Q)$; similarly, for a fixed $d(P^*, Q)$, $P$'s value decreases linearly w.r.t. $\varepsilon$. Importantly, Eq. (2) enables subsequent results and a theoretically justified choice for the reference (e.g., Lemma 4 to derive Theorem 1 in Sec. 4.2).

### 4.1 Data Valuation with a Ground Truth Reference

With Eq. (1), we return to the problem statement described above: given two datasets $D \sim P$ and $D' \sim P'$, under what conditions can we conclude that $\Upsilon(P) > \Upsilon(P')$? We first assume access to a reference dataset $D^* \sim P^*$. We then relax this assumption in Sec. 4.2.

**Proposition 1.** Given datasets $D \sim P$ and $D' \sim P'$, let $m := |D|$ and $m' := |D'|$. Let $D^* \sim P^*$ and $m^* := |D^*|$ be its size. For some bias requirement $\varepsilon_{\text{bias}} \geq 0$ and a required decision margin $\varepsilon_\Upsilon \geq 0$. If $\nu(D) > \nu(D') + \Delta_{\Upsilon,\nu}$ where the *criterion margin* $\Delta_{\Upsilon,\nu} := \varepsilon_\Upsilon + 2[\varepsilon_{\text{bias}} + \sqrt{K/m} + \sqrt{K/m'} + 2\sqrt{K/m^*}]$. Let $\delta := 2\exp(\frac{-\varepsilon_{\text{bias}}^2 \overline{m} m^*}{2K(\overline{m}+m^*)})$ where $\overline{m} = \max\{m, m'\}$. Then, $\Upsilon(P) > \Upsilon(P') + \varepsilon_\Upsilon$ with probability at least $(1 - 2\delta)$.

*(Proof in App. A)* Proposition 1 describes the criterion margin $\Delta_{\Upsilon,\nu}$ such that if $\nu(D) - \nu(D') > \Delta_{\Upsilon,\nu}$—i.e., the criterion is met—we can draw the conclusion that $\Upsilon(P) > \Upsilon(P') + \varepsilon_\Upsilon$ at a confidence level of $1 - 2\delta$. Hence, a smaller $\Delta_{\Upsilon,\nu}$ corresponds to an "easier" criterion to satisfy. The expression $\Delta_{\Upsilon,\nu} = \mathcal{O}(\varepsilon_\Upsilon + \varepsilon_{\text{bias}} + 1/\sqrt{\underline{m}})$ where $\underline{m} := \min\{m, m', m^*\}$ highlights three components that are in tension: a buyer-defined decision margin $\varepsilon_\Upsilon$, a bias requirement $\varepsilon_{\text{bias}}$ from the MMD estimator (Lemma 1), and the minimum size $\underline{m}$ of the vendors' sample datasets (assuming $m^* \geq \max\{m, m'\}$). If the buyer requires a higher decision margin $\varepsilon_\Upsilon$ (i.e., the buyer wants to determine if $P$ is more valuable than $P'$ by a larger margin), then it may be necessary to (i) set a lower bias requirement $\varepsilon_{\text{bias}}$ and/or (ii) request larger sample datasets from the vendors. In (i), suppose $\underline{m}$ remains unchanged, a lower $\varepsilon_{\text{bias}}$ reduces the confidence level $1 - 2\delta$ since $\delta$ increases as $\varepsilon_{\text{bias}}$ decreases. Hence, although the buyer concludes that $P$ is more valuable than $P'$ by a higher decision margin, the corresponding confidence level is lower. In (ii), suppose $\varepsilon_{\text{bias}}$ remains unchanged, a higher minimum sample size $\underline{m}$ increases the confidence level.[2] In other words, to satisfy the buyer's higher decision margin, the

---

[2]In proof of Proposition 1, it is shown that the confidence level strictly increases when $m$ or $m'$ increases.

vendors need to provide larger sample datasets. This can also help the buyer increase their confidence level if the criterion is satisfied. Proposition 1 illustrates the interaction between a buyer and data vendors: The buyer's requirement is represented by the decision margin, and the vendors must provide sufficiently large sample datasets to satisfy this requirement.

## 4.2 Approximating the Reference Distribution

Previously we assumed $P^*$ could be accessed as the reference. We now relax this assumption by replacing $P^*$ with a mixture distribution $P_\omega$ over all the vendors' distributions, as defined in Observation 1. We first prove an error guarantee to generalize Proposition 1 when using $P_\omega$ instead of $P^*$. We then use a game-theoretic formulation to motivate the choice of the *uniform* mixture, $P_\mathrm{U}$.

Formally, using $P_\omega$ as the reference (with $D_\omega \sim P_\omega$) instead of $P^*$ gives the following valuation:

$$\hat{\Upsilon}(P) \coloneqq -d(P, P_\omega), \quad \hat{\nu}(D) \coloneqq -\hat{d}(D, D_\omega). \tag{3}$$

Namely, $\hat{\Upsilon}$ is an approximation to $\Upsilon$ (equiv. $\hat{\nu}$ to $\nu$), with a bounded (approximation) error as follows,

**Proposition 2.** Recall $\varepsilon_\omega, Q_\omega$ from Observation 1. Then, $\forall P, |\Upsilon(P) - \hat{\Upsilon}(P)| \leq \varepsilon_\omega d(Q_\omega, P^*)$.

(*Proof in App. A*) Proposition 2 provides an error bound from using $P_\omega$ as the reference, which linearly depends on $\varepsilon_\omega$ and $Q_\omega$: A lower $\varepsilon_i$ (i.e., $P_i$ has a lower outlier probability) gives a lower $\varepsilon_\omega$, and a lower $d(Q_i, P^*)$ (i.e., $P_i$'s outlier component is closer to $P^*$) leads to a lower $d(Q_\omega, P^*)$, resulting in a smaller error from using $P_\omega$ as the reference. Using this error guarantee, we give our main result, which provides a decision criterion for concluding that for candidate vendor distributions $P$ and $P'$, their valuations satisfy $\Upsilon(P) > \Upsilon(P') + \varepsilon_\Upsilon$, for some user-specified decision margin $\varepsilon_\Upsilon$. Unlike Proposition 1, this result does not require access to ground truth $P^*$, but instead uses a practically-realizable mixture $P_\omega$.

**Theorem 1.** Given datasets $D \sim P$ and $D' \sim P'$, let $m \coloneqq |D|$ and $m' \coloneqq |D'|$. Let $D_\omega, \hat{\nu}$ be from Eq. (3) and $m_N \coloneqq |D_\omega|$. For some bias requirement $\varepsilon_\mathrm{bias} \geq 0$ and a required decision margin $\varepsilon_\Upsilon \geq 0$, suppose $\hat{\nu}(D) > \hat{\nu}(D') + \Delta'_{\Upsilon,\nu}$ where the *criterion margin* $\Delta'_{\Upsilon,\nu} \coloneqq \varepsilon_\Upsilon + 2[\varepsilon_\mathrm{bias} + \sqrt{K/m} + \sqrt{K/m'} + 2\sqrt{K/m_N} + \varepsilon_\omega d(Q_\omega, P^*)]$. Let $\delta' \coloneqq 2\exp(\frac{-\varepsilon_\mathrm{bias}^2 \overline{m} m_N}{2K(\overline{m}+m_N)})$ where $\overline{m} = \max\{m, m'\}$. Then $\Upsilon(P) > \Upsilon(P') + \varepsilon_\Upsilon$ with probability at least $(1 - 2\delta')$.

(*Proof in App. A*) Compared with Proposition 1, the criterion margin $\Delta'_{\Upsilon,\nu}$ has an additional term of $2\varepsilon_\omega d(Q_\omega, P^*)$, which depends on both the size of the outlier component $\varepsilon_\omega$ and the statistical difference $d(Q_\omega, P^*)$ between $Q_\omega$ and $P^*$.[3] This term explicitly accounts for the statistical difference $d(P_\omega, P^*)$ to generalize Proposition 1: $d(P_\omega, P^*) = 0$ recovers Proposition 1. Importantly, this result implies that using $P_\omega$ (to replace $P^*$) retains the previous analysis and interpretation: a buyer's requirement via the decision margin can be satisfied by the vendors providing (sufficiently) large sample datasets, which is empirically investigated in a comparison against existing valuation methods (Sec. 5). We highlight that Theorem 1 exploits the closed property of Huber models (via Observation 1), the triangle inequality of MMD (via Proposition 2) and the uniform convergence of the MMD estimator. Hence, the modeling and design choices of Huber and MMD are both necessary.

### 4.2.1 A Game-theoretic Choice of Mixture

The above results hold for general mixture distributions $P_\omega$, begging the question: Which mixture should one use (i.e., what $\omega$)? A game-theoretic formulation reveals that the uniform strategy is worst-case optimal, so we propose to use the uniform mixture $P_\mathrm{U}$ as the special case of $P_\omega$.

Consider the following two-player zero-sum game. A payoff matrix $\mathcal{R}$ consists of $n$ rows, one corresponding to each vendor index, and $n!$ columns, one corresponding to each permutation over the vendor indices. The row player (i.e., the agent conducting the data valuation) picks a vendor index $r \in N$. The column player (hypothetical adversary) then adversarially chooses a permutation $\pi_c$ over the indices in $N$. Hence, the action space for the row player is $N$ and that for the column player is all possible permutations of $\{1, 2 \ldots, n\}$.[4] The column player represents the fact that the row player

---

[3]Compared with $\Delta_{\Upsilon,\nu}$ in Proposition 1, if $m^* = m_N$, then $\Delta'_{\Upsilon,\nu} = \Delta_{\Upsilon,\nu} + 2\varepsilon_\omega d(Q_\omega, P^*)$.

[4]W.l.o.g. assume a fixed ordering of all $n!$ permutations.

lacks prior knowledge about vendors: hence it selects an index $r$ in any possible arbitrary permutation $\pi_c$. Then, for a pair of actions $(r, \pi_c)$, the quality of the distribution $P_{\pi_c[r]}$ is the row player's payoff $\mathcal{R}_{r,c} := -d(P^*, P_{\pi_c[r]})$, defined as the negated MMD between $P_{\pi_c[r]}$ and the optimal distribution $P^*$ (i.e., a lower MMD means a higher payoff), specifying the following optimization:

$$\max_r \min_c \mathcal{R}_{r,c} \tag{4}$$

where $\mathcal{R} \in \mathbb{R}^{n \times (n!)}$ is the payoff matrix and $\max$ ($\min$) denotes the row (column) player's action. A strategy $s_{\text{row}} \in \triangle(n-1)$ (as an $n$-dimensional probability vector) specifies the probability with which the row player picks a data vendor (at a position).

While efficient linear program solvers to Eq. (4) are available for explicitly specified $\mathcal{R}$, in our setting, $\mathcal{R}$ is not explicitly specified due not knowing $P^*$. Fortunately, we show that the uniform strategy is optimal *without* knowing $\mathcal{R}$ explicitly:

**Proposition 3.** The optimal solution for the row player to Eq. (4) is $s_{\text{row}}^* = \left[\frac{1}{n}, \frac{1}{n}, \dots, \frac{1}{n}\right]$.

(*Proof in App. A*) Intuitively, a uniform strategy over the vendors cannot be exploited by the column player, and is thus worst-case optimal. We highlight that while uniform strategy being worst-case optimal may seem intuitive, the mathematical properties and derivations needed are less straightforward. In particular, the proof depends on the "closed" property of Huber (i.e., Observation 1) and the "linearity" of MMD applied to Huber (i.e., Eq. (2)) to exploit the strong duality of a linear program. We also discuss alternative formulations to Eq. (4) in App. A.

Then, we adopt the uniform mixture $P_{\text{U}}$ as the special case of $P_\omega$ in Eq. (3):

$$\hat{\Upsilon}(P) = -d(P, P_{\text{U}}), \quad \hat{\nu}(D) := -\hat{d}(D, D_{\text{U}}). \tag{5}$$

Proposition 2 and Theorem 1 are applied directly in App. A. The uniform mixture $P_{\text{U}}$ is inspired from the solutions to Eq. (4), which is a game based on *not* having prior knowledge about the vendors. In this setting of no ground truth and no prior knowledge about the vendors, one might wonder if/how we can derive a lower bound of error to crystalize the difficulty of the problem (our Proposition 2 gives an upper bound of error for general $P_\omega$, and Corollary 1 is for $P_{\text{U}}$). We expand this in App. C, making references to robust statistics and mechanism design.

## 5 Empirical Results

We first compare the sample efficiency of several baselines, and then investigate the effectiveness of our method in ranking $n$ data distributions. Additional information on experimental settings is in App. C and additional results under non-Huber settings, and on scalability are in App. D. Our code is available at https://github.com/XinyiYS/Data_Distribution_Valuation.

**Baselines.** To accommodate the existing methods which explicitly require a validation set (Sec. 2), we perform some experiments using a validation set $D_{\text{val}} \sim P^*$. This assumption is made only for empirical comparison, and subsequently relaxed. The baselines that explicitly require $D_{\text{val}}$ are class-wise Shapley (CS) [55, Eq. (3)], LAVA [33] and DAVINZ [67, Eq. (3)]; the baselines that do not require $D_{\text{val}}$ are information-gain value (IG) [56, Eq. (1)], volume value (VV) [70, Eq. (2)] and MMD$^2$ [60, Eq. (1)], which implements a biased estimator of MMD$^2$. For each baseline, we adopt their official implementation if available. Note that though theoretically MMD$^2$ is obtained by squaring MMD, the estimator for MMD$^2$ is *not* obtained by squaring the MMD estimator (elaborated in App. B), so they give different results. Note that DAVINZ also includes the MMD as a specific implementation choice, linearly combined with a neural tangent kernel (NTK)-based score. However, their theoretical results are specific to NTK and not MMD, while our result (e.g., Theorem 1) is MMD-specific.

Our implementation of MMD, including the radial basis function kernel, follows [42]. To implement our proposed uniform mixture $P_{\text{U}}$ in cases where $D_i$'s have different sizes, we do the following: denote the minimum dataset size by $m_{\text{min}} := \min_i |D_i|$. Then for each $D_i$, uniformly randomly sample a subset $D_{i,\text{sub}} \subseteq D_i$ of size $m_{\text{min}}$ from $D_i$, and use the union $D_{\text{U}} := \cup_i D_{i,\text{sub}}$.

**Datasets.** We consider both classification (Cla.) and regression (Reg.) since some baselines (i.e., CS, LAVA) are specific to classification while some (i.e., IG, VV) are specific to regression. Our method is applicable to both. CaliH (resp. KingH) is a housing prices dataset in California [35] (resp. in

Kings county [28]). Census15 (resp. Census17) is a personal income prediction dataset from the 2015 (resp. 2017) US census. [48]. Credit7 [49] and Credit31 [3] are two credit card fraud detection datasets. TON [47] and UGR16 [45] are two network intrusion detection datasets.

Many of our evaluations are conducted under a Huber model, which requires matched supports of $P^*$ and $Q$, such as MNIST, EMNIST and FaMNIST, all in $\mathbb{R}^{32 \times 32}$, CIFAR10 and CIFAR100, and Census15 and Census17. Other datasets require additional pre-processing: CaliH and KingH are standardized and pre-processed separately to be in $\mathbb{R}^{10}$. Additional pre-processing details in App. D. Subsequently, each $P_i$ follows a Huber: $P_i = (1 - \varepsilon_i)P^* + \varepsilon_i Q$ (i.e., $\forall i, Q_i = Q$). We also run experiments on non-Huber settings in App. D, where our method remains effective.

**ML model M.** For model-specific baselines such as DAVINZ and CS, in Sec. 5.2, we adopt a 2-layer convolutional neural network (CNN) for MNIST, EMNIST, FaMNIST; ResNet-18 [29] for CIFAR10 and CIFAR100; logistic regression (LogReg) for Credit7 and Credit31, and TON and UGR16; linear regression (LR) for CaliH and KingH, and Census15 and Census17. Details are in App. D.

## 5.1 Sample Efficiency via Empirical Convergence

Our goal is to find a sample-efficient policy that correctly compares $\Upsilon(P)$ vs. $\Upsilon(P')$ by comparing $\nu(D)$ vs. $\nu(D')$, even if the sizes of $D, D'$ are small. In practice, there is no direct access to $P, P'$ but only $D, D'$; the sizes of $D, D'$ may not be very large. Here, we compare $\{\nu(D_i)\}_{i \in N}$ to $\{\Upsilon(P_i)\}_{i \in N}$ as we vary dataset size $m_i$.

**Setting.** We implement $\Upsilon(P_i)$ as approximated by a $\nu(D_i^*)$ where $D_i^* \sim P_i$ with a large size $m_i^*$ (e.g., $10,000$ for $P^* = $ MNIST vs. $Q = $ EMNIST). Denote the values of the samples as $\boldsymbol{\nu}_{m_i} := \{\nu(D_i)\}_{i \in N}$ where the sample size $m_i = |D_i|$ and the approximated ground truths as $\boldsymbol{\nu}^* := \{\nu(D_i^*)\}_{i \in N}$; in this way, $\boldsymbol{\nu}^*$ is well-defined respectively for each comparison baseline (i.e., *not* our MMD definition Eq. (1)). We highlight that each $\nu$ (i.e., each baseline) is evaluated against its corresponding $\boldsymbol{\nu}^*$ to demonstrate the empirical convergence. This is to examine the practical applicability of each $\nu$ when the sizes of the provided $\{D_i\}_{i \in N}$ are limited.

**Evaluation and results.** We evaluate three criteria—the $\ell_2$ and $\ell_\infty$ errors and the number of pair-wise inversions as follows, $\|\boldsymbol{\nu}_{m_i} - \boldsymbol{\nu}^*\|_2, \|\boldsymbol{\nu}_{m_i} - \boldsymbol{\nu}^*\|_\infty$ and $\texttt{inversions}(\boldsymbol{\nu}_{m_i}, \boldsymbol{\nu}^*) := (1/2) \sum_{i,i' \in [n], i \neq i'} \mathbb{1}(\nu(D_i^*) > \nu(D_{i'}^*) \wedge \nu(D_i) < \nu(D_{i'}))$. In words, if the conclusion via $\nu(D_i)$ vs. $\nu(D_{i'})$ *differs* from that via $\nu(D_i^*)$ vs. $\nu(D_{i'}^*)$, it is an inversion. For all 3 criteria, lower is better.

Fig. 1 (and Figs. 2 to 5 in App. D) demonstrate that our MMD-based method is overall (one of) the *most sample-efficient* for different evaluation criteria and datasets, validating the theoretical results (Table 1) that MMD is more sample-efficient than WD.

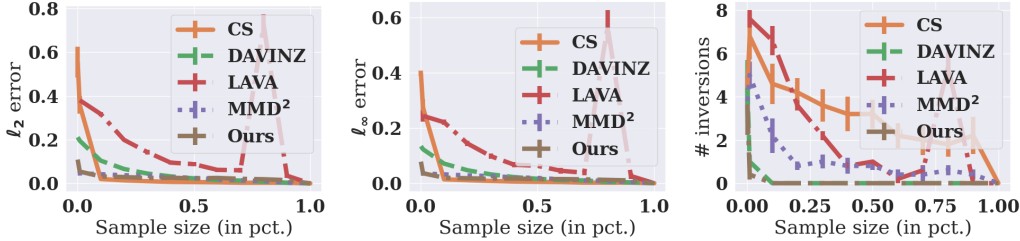

Figure 1: The 3 criteria (on $y$-axis) for $P^* = $ MNIST vs. $Q = $ EMNIST. $n = 5, m_i^* = 10,000$. $x$-axis shows sample size in percentage, i.e., $m_i/m_i^*$ where $m_i^*$ is fixed to investigate how the criteria change w.r.t. $m_i/m_i^*$: If the criteria decrease quickly w.r.t. $m_i/m_i^*$, it means the metric converges quickly (i.e., sample-efficient). Results averaged (standard errors bars) over 5 independent trials.

## 5.2 Ranking Data Distributions

Motivated by the use-case of a buyer identifying the best data vendor(s), we measure our ability to rank $n$ distributions based on the values of sample datasets.

**Setting.** For a valuation metric $\nu$ (e.g., Eq. (1)), denote the values of datasets from all vendors as $\boldsymbol{\nu} := \{\nu(D_i)\}_{i \in N}$. To compare against different baselines (i.e., other definitions of $\nu$), we define the

following *common* ground truth, the expected test performance $\zeta_i \coloneqq \mathbb{E}_{D_i \sim P_i}[\mathrm{Perf}(\mathbf{M}(D_i); D_{\text{test}})]$ of an ML model $\mathbf{M}(D_i)$ trained on $D_i$, over a fixed test set $D_{\text{test}}$ where the expectation is over the randomness of $D_i$. Let $\boldsymbol{\zeta} \coloneqq \{\zeta_i\}_{i \in N}$. The ML model $\mathbf{M}$ is specified previously and the test set $D_{\text{test}}$ is from the respective $P^*$ and *not* seen by any data vendor. Note that $D_{\text{test}}$ is used to obtain $\zeta_i$ for comparison purposes, and it is *not* to be confused with $D_{\text{val}}$, which is required by some baselines as part of their methods.

**Utilizing labels.** We also extend our method to explicitly consider the label information via the conditional distributions of labels given features (i.e., $P_{Y|X}$), denoted as *Ours cond.* Other baselines such as LAVA and CS already explicitly use label information. Specifically, for $D_i$ containing paired features and labels, we fit a learner $\mathbf{M}(D_i)$ on $D_i$ and use its predictions on $D_{\text{val}}$ (thus we require $D_{\text{val}}$) as an empirical representation of the $P_{Y|X}$ for $D_i$ and compute the MMD between the conditional distributions (more implementation details in App. D). Unlike our original method (i.e., Ours), this variant differs in exploiting the feature-label pairs in $D_i$. We relax the assumption $D_{\text{val}} \sim P^*$ by replacing $D_{\text{val}}$ with $D_{\text{U}}$, namely for the baselines needing an explicit reference, we use $D_{\text{U}}$. The resulting data values are denoted as $\hat{\boldsymbol{\nu}}$ (the data values based on $D_{\text{val}}$ are denoted as $\boldsymbol{\nu}$).

**Evaluation metric.** Here $\nu$ is effective if it identifies the most valuable sampling distribution, or more generally, if $\boldsymbol{\nu}$ preserves the ranking of $\boldsymbol{\zeta}$. In other words, the ranking of the data vendors' sampling distributions $\{P_i\}_{i \in N}$ is correctly identified by the values of their datasets $\{D_i\}_{i \in N}$, quantified via the Pearson correlation coefficient: $\rho(\boldsymbol{\nu}, \boldsymbol{\zeta})$ (higher is better). Note that we compare the rankings of different baselines instead of the actual data values which can be on different scales.

**Results.** We report the average and standard error over 5 independent random trials, on CIFAR10/CIFAR100, TON/UGR16, CaliH/KingH and Census15/Census17 in Tables 2 and 3 respectively, and defer the others to App. D. Note that when $D_{\text{val}}$ is unavailable (i.e., right columns), Ours cond. is not applicable because the label-feature pair information is not well-defined under the Huber model (e.g., for $P^* = \mathrm{CIFAR10}$ vs. $Q = \mathrm{CIFAR100}$, the label of a CIFAR100 image is not well-defined for a model trained for CIFAR10). The $\mathrm{Perf}(\cdot)$ for $\boldsymbol{\zeta}$ for classification (resp. regression) is accuracy (resp. coefficient of determination (COD)), so higher is better.

Table 2: Pearson correlations between data sample values and data distribution values for classification.

| Baselines | CIFAR10 vs. CIFAR100 | | TON vs. UGR16 | |
| | $\rho(\boldsymbol{\nu}, \boldsymbol{\zeta})$ | $\rho(\hat{\boldsymbol{\nu}}, \boldsymbol{\zeta})$ | $\rho(\boldsymbol{\nu}, \boldsymbol{\zeta})$ | $\rho(\hat{\boldsymbol{\nu}}, \boldsymbol{\zeta})$ |
| --- | --- | --- | --- | --- |
| LAVA | -0.907(0.01) | -0.924(0.01) | 0.254(0.26) | -0.159(0.38) |
| DAVINZ | -0.437(0.10) | -0.481(0.13) | -0.201(0.26) | -0.529(0.21) |
| CS | 0.889(0.03) | -0.874(0.02) | 0.451(0.19) | 0.256(0.28) |
| MMD$^2$ | 0.764(0.02) | 0.563(0.01) | 0.526 (0.11) | **0.480(0.15)** |
| Ours | 0.763(0.02) | **0.564(0.02)** | **0.584(0.17)** | 0.461(0.14) |
| Ours cond. | **0.989(0.01)** | N.A. | 0.562(0.16) | N.A. |

For classification, Table 2 shows that our method performs well when $D_{\text{val}}$ is available (e.g., Ours cond. is the highest for CIFAR10 vs. CIFAR100 under $\rho(\boldsymbol{\nu}, \boldsymbol{\zeta})$) and also when $D_{\text{val}}$ is unavailable (e.g., Ours as highest for CIFAR10 vs. CIFAR100 under $\rho(\hat{\boldsymbol{\nu}}, \boldsymbol{\zeta})$). MMD$^2$ performs comparably to Ours, which is expected since in theory their values differ only by a square and the evaluation mainly focuses on the rank, instead of the absolute values. We also note that CS, by exploiting the label information in classification, performs competitively with $D_{\text{val}}$, but performs sub-optimally without $D_{\text{val}}$. This is because the label information in $D_{\text{val}}$ is no longer available in $D_\omega$ (due to $D_\omega$ being Huber). LAVA and DAVINZ, both exploiting the gradients of the ML model, do not perform well. The reason could be that under the Huber model, the gradients are not as informative about the values of the data. Intuitively, while the gradient of (the loss of) a data point on an ML model can be informative about the value of this data point, this reasoning is not applicable here, because the data point may not be from the same true distribution $P^*$: The value of a gradient obtained on a CIFAR100 image to an ML model intended for CIFAR10 may not be informative about the value of this CIFAR100 image. We highlight that neither of LAVA and DAVINZ was originally proposed for such cases (i.e., the Huber model).

Table 3: Pearson correlations between data sample values and data distribution values for regression.

| Baselines | CaliH vs. KingH | | Census15 vs. Census17 | |
|---|---|---|---|---|
| | $\rho(\boldsymbol{\nu}, \boldsymbol{\zeta})$ | $\rho(\hat{\boldsymbol{\nu}}, \boldsymbol{\zeta})$ | $\rho(\boldsymbol{\nu}, \boldsymbol{\zeta})$ | $\rho(\hat{\boldsymbol{\nu}}, \boldsymbol{\zeta})$ |
| IG | -0.907(0.02) | | -0.932(0.02) | |
| VV | -0.603(0.01) | | -0.707(0.01) | |
| DAVINZ | 0.852(0.03) | 0.048(0.08) | 0.779(0.14) | 0.227(0.11) |
| MMD$^2$ | 0.872(0.03) | 0.726(0.09) | **0.889(0.05)** | **0.838(0.08)** |
| Ours | **0.896(0.02)** | **0.767(0.04)** | 0.843(0.03) | 0.769(0.08) |
| Ours cond. | 0.812(0.02) | N.A. | 0.848(0.06) | N.A. |

For regression, Table 3 shows that Ours and MMD$^2$ continue to perform well while baselines (i.e., IG and VV) that completely remove the reference perform poorly, as they cannot account for the statistical heterogeneity without a reference. Notably, DAVINZ performs competitively for when $D_{\text{val}}$ is available, due to its implementation utilizing a linear combination of an NTK-based score (i.e., gradient information) and MMD (similar to Ours), via an auto-tuned weight between the two. We find that for classification, the NTK-based score is dominant while for regression (and available $D_{\text{val}}$) the MMD is dominant. This could be because the models are more complex for the classification tasks (e.g., ResNet-18) as compared to linear regression models for regression, so the obtained gradients are more significant (i.e., higher numerical NTK-based scores). Thus, for regression, DAVINZ produces values similar to Ours, hence the similar performance. We highlight that DAVINZ focuses on the use of NTK w.r.t. a given reference, while our method focuses on MMD *without* such a reference, as evidenced by Ours outperforming DAVINZ without $D_{\text{val}}$ (i.e., the columns under $\rho(\hat{\boldsymbol{\nu}}, \boldsymbol{\zeta})$ in Table 3).

## 6 Discussion

Under a Huber model of vendor heterogeneity, we propose an MMD-based data distribution valuation and derive theoretically-justified policies for comparing distributions from their respective samples. To address the lack of access to the true reference distribution, we use a convex mixture of the vendors' distributions as the reference, and derive a corresponding error guarantee and comparison policy. Then, we specifically select the uniform mixture as a game-theoretic choice when no prior knowledge about the vendors is assumed. Empirical results demonstrate that our method performs well in efficiently identifying the most valuable data distribution. While our theoretical results are limited to the Huber model, MMD is observed to be effective under two non-Huber settings (App. D.3.4). Extending the theory to more general heterogeneity models is an interesting direction for future study.

## Acknowledgments and Disclosure of Funding

Xinyi Xu is supported by the Institute for Infocomm Research of Agency for Science, Technology and Research (A*STAR) and would like to thank Peiyu Hu and Siyu Liu for their support. This research/project is supported by the National Research Foundation Singapore and DSO National Laboratories under the AI Singapore Programme (AISG Award No: AISG2-RP-2020-018). Giulia Fanti and Shuaiqi Wang were supported in part by National Science Foundation grant CCF-2338772, as well as the Sloan Foundation, Bosch, and Intel.

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

# A    Proofs and Derivations

## A.1    Equivalent Definitions of MMD

Definition 1 follows [26] to use the function class $\mathcal{F}$. In our derivation and implementations, we adopt an equivalent but more convenient form utilizing the kernel and mean embedding [26, Section 2.2]. Definition 1 is more difficult to use with because of the sup operation. Specifically, recall that $\mathcal{F}$ is a class of functions in the unit ball of reproducing kernel Hilbert space (RKHS) $\mathcal{H}$ which is associated with the kernel $k$, then the following equivalence holds [26, Lemma 6]:

$$d(P, P'; \mathcal{F}) = \{\mathbb{E}_{X,X'\sim P}[k(X, X')] - 2\mathbb{E}_{X\sim P, Y\sim P'}[k(X, Y)] + \mathbb{E}_{Y,Y'\sim P'}[k(Y, Y')]\}^{1/2} \ . \quad (6)$$

The definition of MMD that is used in our derivation and implementation is the right hand side above. This definition is more convenient because it no longer has the unwieldy sup operation and is interpretable (i.e., resembling the square root of an expanded quadratic expression).

We note that throughout the paper, the class $\mathcal{F}$ of functions, its corresponding RKHS $\mathcal{H}$ and the associated kernel $k$ are all kept constant, so their notational dependence is omitted where clear for brevity.

## A.2    For Sec. 3

**Proof of Observation 1.**

*Proof of Observation 1.*  Note that $P_\omega$ is a mixture model/distribution, as it is a convex combination of $P_i$'s. The expressions for $\varepsilon_N, Q_N$ are derived as follows,

$$\begin{aligned}
P_\omega &:= \sum_{i\in N} \omega_i[(1-\varepsilon_i)P^* + \varepsilon_i Q_i] \\
&= P^* \sum_{i\in N} \omega_i(1-\varepsilon_i) + (\sum_{i\in N} \omega_i\varepsilon_i Q_i) \\
&= P^*(1 - \underbrace{\sum_{i\in N} \omega_i\varepsilon_i}_{\varepsilon_\omega}) + (\underbrace{\sum_{i\in N} \omega_i\varepsilon_i Q_i}_{\varepsilon_\omega Q_\omega}) \ .
\end{aligned}$$

The last step uses the fact that the entries of a probability vector $\omega$ sum up to $1$ .     □

## A.3    For Sec. 4.1

**Useful lemma.**

**Lemma 1** (Uniform Convergence of MMD Estimator [26, Theorem 7]). Let $X \sim P, W \sim P'$ and the size of $X$ is $m$, the size of $W$ is $n$. Then the biased MMD estimator $\hat{d}$ satisfies the following approximation guarantee:

$$\Pr_{X,W}\left\{|\hat{d}(X, W) - d(P, P')| > 2(\sqrt{K/m} + \sqrt{K/n}) + \varepsilon\right\} \le 2\exp(\frac{-\varepsilon^2 mn}{2K(m+n)})$$

where $\Pr_{X,W}$ is over the randomness of the $m$-sample $X$ and $n$-sample $W$.

Note that $\varepsilon_{\text{bias}}$ in our results (e.g., Proposition 1) corresponds to $\varepsilon$ in Lemma 1.

**Proof of Proposition 1.**

*Proof of Proposition 1.*  Apply Lemma 1 to $D \sim P$ and $D \sim P'$, respectively.

W.p. $\ge 1 - \delta_P$ where $\delta_P := 2\exp(\frac{-\varepsilon_{\text{bias}}^2 mm^*}{2K(m+m^*)})$,

$$d(P, P^*) \le \hat{d}(D, D^*) + [2(\sqrt{\frac{K}{m}} + \sqrt{\frac{K}{m^*}}) + \varepsilon_{\text{bias}}]$$

$$-\Upsilon(P) \leq -\nu(D) + [2(\sqrt{\frac{K}{m}} + \sqrt{\frac{K}{m^*}}) + \varepsilon_{\text{bias}}]$$

$$\Upsilon(P) \geq \underbrace{\nu(D) - [2(\sqrt{\frac{K}{m}} + \sqrt{\frac{K}{m^*}}) + \varepsilon_{\text{bias}}]}_{A}$$

where the first inequality is from directly applying Lemma 1, the second inequality is from substituting the definitions in Eq. (1).

Symmetrically, w.p. $\geq 1 - \delta_{P'}$ where $\delta_{P'} := 2\exp(\frac{-\varepsilon_{\text{bias}}^2 m' m^*}{2K(m'+m^*)})$,

$$\Upsilon(P') \leq \underbrace{\nu(D') + [2(\sqrt{\frac{K}{m'}} + \sqrt{\frac{K}{m^*}}) + \varepsilon_{\text{bias}}]}_{B} .$$

Observe that if $A \geq B + \varepsilon_\Upsilon$, then apply the independence assumption (between $D \sim P$ and $D' \sim P'$), w.p. $\geq (1 - \delta_P)(1 - \delta_{P'})$, $\Upsilon(P) > \Upsilon(P') + \varepsilon_\Upsilon$ .

Re-arrange the terms in $A \geq B + \varepsilon_\Upsilon$ to derive $\zeta_\nu$,

$$\nu(D) - [2(\sqrt{\frac{K}{m}} + \sqrt{\frac{K}{m^*}}) + \varepsilon_{\text{bias}}] \geq \nu(D') + [2(\sqrt{\frac{K}{m'}} + \sqrt{\frac{K}{m^*}}) + \varepsilon_{\text{bias}}] + \varepsilon_\Upsilon$$

$$\nu(D) \geq \nu(D') + \underbrace{2[\varepsilon_{\text{bias}} + \sqrt{\frac{K}{m}} + \sqrt{\frac{K}{m'}} + 2\sqrt{\frac{K}{m^*}}]}_{\zeta_\nu} + \varepsilon_\Upsilon .$$

To arrive at the simpler but slightly looser result in the main paper. Note that
$$(1 - \delta_P)(1 - \delta_{P'}) \geq (1 - \delta)^2 \geq (1 - 2\delta) .$$

**Confidence level increases with $m, m'$.** Note that equivalently,
$$\delta_P = 2\exp(-\frac{\varepsilon_{\text{bias}}^2 m^*}{2K} + \frac{\varepsilon_{\text{bias}}^2 m^{*2}}{2K(m+m^*)}) ,$$

which is decreasing in $m$. Similarly for $\delta_{P'}$ w.r.t. $m'$. As a result, a higher $m$ implies a lower $\delta_P$ and thus a higher confidence level $(1 - \delta_P)(1 - \delta_{P'})$.

$\square$

## A.4    For Sec. 4.2

**Useful lemma.**

**Lemma 2** ([15, In proof of Lemma 3.3]). For a Huber model $P := (1 - \varepsilon)P^* + \varepsilon Q$, the MMD $d(P, P^*) = \varepsilon d(P^*, Q)$.

### A.4.1    Results and Discussion for a General Mixture $P_\omega$

**Error from using a reference.** Sec. 4.2 describes a theoretically justified choice for a reference $P_{\text{ref}}$ to be used in place of $P^*$ in $\Upsilon$ in Eq. (1), and define an approximate $\hat{\Upsilon} := -d(P_{\text{ref}}, P)$. Then, the valuation error $|\Upsilon(P) - \hat{\Upsilon}(P)|$ from using this reference $P_{\text{ref}}$ should ideally be small. This error is directly upper bounded by the MMD between $P_{\text{ref}}$ and $P^*$, as follows.

**Lemma 3.** For a choice of reference $P_{\text{ref}}$, define $\hat{\Upsilon}(P) := -d(P_{\text{ref}}, P)$ as the approximate version of $\Upsilon(P)$ in Eq. (1). Then,
$$\forall P, |\Upsilon(P) - \hat{\Upsilon}(P)| \leq d(P_{\text{ref}}, P^*) .$$

*Proof.* Apply the triangle inequality of MMD to the definitions of $\Upsilon$ and $\hat{\Upsilon}$.
$$|\Upsilon(P) - \hat{\Upsilon}(P)| = |d(P, P^*) - d(P_{\text{ref}}, P^*)|$$
$$\leq d(P, P_{\text{ref}}) .$$

$\square$

Lemma 3 implies that a better reference distribution (i.e., with a lower MMD to $P^*$) leads to a better approximate $\hat{\Upsilon}$ (i.e., with a lower error). Hence, we have specifically obtained theoretical results to upper bound the MMD between our considered choices for reference: $P_\omega$ via Lemma 4, which is be directly combined with Lemma 3 to derive the error guarantee, described next.

**Error from using $P_\omega$ as the reference.** We first provide an upper bound on $d(P_\omega, P^*)$ in Lemma 4, and then combine it with Lemma 3 to obtain Proposition 2.

**Lemma 4.** The sampling distribution $P_\omega$ for $D_\omega$ satisfies $d(P_\omega, P^*) \leq \varepsilon_\omega d(Q_\omega, P^*)$ .

*Proof of Lemma 4.* The proof is a direct application of Observation 1 and Lemma 2. □

Lemma 4 provides an upper bound on the MMD between $P_\omega$ and $P^*$, which linearly depends on $\varepsilon_\omega$ and $Q_\omega$. A lower $\varepsilon_\omega$ or a lower $d(Q_\omega, P^*)$ leads to a smaller $d(P_\omega, P^*)$, making $D_\omega$ a better reference.

*Proof of Proposition 2.* It directly combines Lemma 4 and Lemma 3. □

*Proof of Theorem 1.* Apply Lemma 4 and Lemma 1 to $D \sim P$ and $D \sim P'$, respectively.

W.p. $\geq 1 - \delta'_P$ where $\delta'_P := 2\exp(\frac{-\varepsilon_{\text{bias}}^2 m m_N}{2K(m+m_N)})$,

$$\Upsilon(P) = -d(P, P^*) \geq -d(P, P_\omega) - d(P_\omega, P^*) \geq \underbrace{\hat{\nu}(D) - [2(\sqrt{\frac{K}{m}} + \sqrt{\frac{K}{m_N}}) + \varepsilon_{\text{bias}}] - \varepsilon_\omega d(Q_\omega, P^*)}_{A} ,$$

symmetrically, w.p. $\geq 1 - \delta'_{P'}$ where $\delta'_{P'} := 2\exp(\frac{-\varepsilon_{\text{bias}}^2 m' m_N}{2K(m'+m_N)})$,

$$\Upsilon(P') = -d(P', P^*) \leq -d(P', P_\omega) + d(P_\omega, P^*) \leq \underbrace{\hat{\nu}(D') + [2(\sqrt{\frac{K}{m'}} + \sqrt{\frac{K}{m_N}}) + \varepsilon_{\text{bias}}] + \varepsilon_\omega d(Q_\omega, P^*)}_{B} .$$

Observe that if $A \geq B + \varepsilon_\Upsilon$, and apply the independence assumption (between $D \sim P$ and $D' \sim P'$), then w.p. $\geq (1 - \delta'_P)(1 - \delta'_{P'})$, $\Upsilon(P) > \Upsilon(P') + \varepsilon_\Upsilon$ .

Re-arrange the terms in $A \geq B + \varepsilon_\Upsilon$ to derive $\zeta'_\nu$,

$$\hat{\nu}(D) - [2(\sqrt{\frac{K}{m}} + \sqrt{\frac{K}{m_N}}) + \varepsilon_{\text{bias}}] - \varepsilon_\omega d(Q_\omega, P^*) \geq \hat{\nu}(D') + [2(\sqrt{\frac{K}{m'}} + \sqrt{\frac{K}{m_N}}) + \varepsilon_{\text{bias}}] + \varepsilon_\omega d(Q_\omega, P^*) + \varepsilon_\Upsilon$$

$$\hat{\nu}(D) \geq \hat{\nu}(D') + 2\underbrace{\left[\varepsilon_{\text{bias}} + \sqrt{\frac{K}{m}} + \sqrt{\frac{K}{m'}} + 2\sqrt{\frac{K}{m_N}} + \varepsilon_\omega d(Q_\omega, P^*)\right]}_{\zeta'_\nu} + \varepsilon_\Upsilon .$$

To arrive at the simpler but slightly looser result in the main paper, note that

$$(1 - \delta'_P)(1 - \delta'_{P'}) \geq (1 - \delta')^2 \geq (1 - 2\delta') .$$

□

### A.4.2 Results and Discussion for the Uniform Mixture $P_U$

As we do *not* make any assumptions about the prior knowledge of the vendors (e.g., some vendor $i$ is more reputable), we adopt a perspective that minimizes the worst-case (i.e., maximum) error over selecting the vendor's distribution as the reference. In other words, we (represented below as the row player) want to pick a vendor, and use the corresponding distribution as the reference to value other distributions. Hence, if we pick a "good" vendor (i.e., whose distribution is close to $P^*$), it is more desirable (i.e., a higher payoff) since the approximation error (as in Proposition 2) is lower. Formally, we construct a finite two-player zero-sum game in which we show that the uniform strategy is worst-case optimal. Hence, we propose to use the uniform mixture $P_U = \frac{1}{n}\sum_i P_i$ of the vendor's distributions as the specific choice of reference.

**A finite, two-player zero-sum game.** The row player (main player) represents the buyer (the one performing valuation) and the column player (hypothetical adversary) is used to explicitly model the fact that we do *not* have prior knowledge or assumptions about the vendors' distributions (except each $P_i$ is a Huber).

Recall that the payoff matrix $\mathcal{R} \in \mathbb{R}^{n \times (n!)}$ for the row player (main player) is

$$\mathcal{R}_{r,c} := -d_{\pi_c[r]} := -d(P^*, P_{\pi_c[r]}), \tag{7}$$

which is the negated MMD between $P^*$ and the distribution $P_{\pi_c[r]}$ at the $r$-th position of the permutation $\pi_c$ and hence the payoff matrix $\mathcal{R}$ is

$$\mathcal{R} := \begin{bmatrix} -d_{\pi_1[1]} & -d_{\pi_2[1]} & -d_{\pi_3[1]} & \cdots & -d_{\pi_{n!}[1]} \\ -d_{\pi_1[2]} & -d_{\pi_2[2]} & -d_{\pi_3[3]} & \cdots & -d_{\pi_{n!}[3]} \\ \vdots & \vdots & \vdots & \ddots & \vdots \\ -d_{\pi_1[n]} & -d_{\pi_2[n]} & -d_{\pi_3[n]} & \cdots & -d_{\pi_{n!}[n]} \end{bmatrix}.$$

The notation $r$ is used specifically refer to the position in a permutation $\pi_c$, and is not to be confused with indexing $i$ of the data vendors. For completeness, the payoff matrix $\mathcal{C}$ of the column player is the negation of $\mathcal{R}$ (i.e., $\mathcal{R} + \mathcal{C} = \mathbf{0}$). Observe that this is a finite, two-player zero-sum game.

**Proposition 3: The uniform strategy is worst-case optimal.** While the values of the entries in $\mathcal{R}$ are not known explicitly, there are some properties of these values that can be exploited. We provide a constructive proof: formulate the primal and dual linear programs (LPs) for the two-player game as in Eq. (4), show that the uniform strategies lead to equal values for both LPs, and conclude that, by the strong duality, both are optimal.

*Proof of Proposition 3.* For the row player, the optimal strategy $s_{\text{row}}^*$ can be computed via the following LP (1) [18], the value of which is equal to the row player's payoff based on the column player's optimal response $s_{\text{col}}^*$ to $s_{\text{row}}^*$:

$$\begin{aligned} \max \ & z \\ \text{s.t.} \ & s_{\text{row}}^\top \mathcal{R} \succcurlyeq z\mathbf{1}^\top \\ & s_{\text{row}}^\top \mathbf{1} = 1 \\ & s_{\text{row}} \succcurlyeq \mathbf{0}. \end{aligned} \qquad \text{LP (1)}$$

where $\succcurlyeq$ denotes element-wise $\geq$.

The dual LP of LP (1) is

$$\begin{aligned} \min \ & z' \\ \text{s.t.} \ & -s_{\text{col}}^\top \mathcal{R}^\top + z'\mathbf{1}^\top \succcurlyeq \mathbf{0} \\ & s_{\text{col}}^\top \mathbf{1} = 1 \\ & s_{\text{col}} \succcurlyeq \mathbf{0}. \end{aligned} \qquad \text{LP (2)}$$

By a change of variable $z'' = -z'$ and the fact that $\mathcal{C} = -\mathcal{R}$, we equivalently rewrite LP (2) as follows,

$$\begin{aligned} \max \ & z'' \\ \text{s.t.} \ & \mathcal{C} s_{\text{col}} \succcurlyeq z''\mathbf{1} \\ & s_{\text{col}}^\top \mathbf{1} = 1 \\ & s_{\text{col}} \succcurlyeq \mathbf{0}. \end{aligned} \qquad \text{LP (3)}$$

Observe that LP (3) is the LP for the column player to solve for $s_{\text{col}}^*$ [18].

Our proof has the main following steps:

1. Verify that the uniform strategy $[1/n, 1/n, \ldots, 1/n]$ is a feasible solution to LP (1) and obtain the corresponding value $z$.

2. Verify that the uniform strategy $[1/(n!), 1/(n!), \ldots, 1/(n!)]$ is a feasible solution to LP (3), obtain the corresponding value $z''$, and translate to the value $z' = -z''$ to LP (2).

3. Apply the *strong duality*: If the values of the primal and dual LPs are equal (i.e., $z = z'$), then the corresponding solutions must both be optimal.

For step 1., we make use of a key observation that the set $\mathcal{G}$ of distributions is fixed, which implies that *independent of* the permutations $\pi_c$, and the sum $S := \sum_{r=1}^{n} -d(\pi_c[r])$ is thus a constant. Then, for any $\pi_c$, it gives the $z_{\pi_c} = \frac{S}{n}$, so the overall value is $z := \min_c z_{\pi_c} = S/n$.

For step 2., in the $r$-th row of $\mathcal{C}$:

$$[d_{\pi_1[r]}, d_{\pi_2[r]}, \ldots, d_{\pi_{n!}[r]}],$$

there are $(n-1)!$ entries/copies of each $d_i$, $1 \leq i \leq n$. This is because out of $n!$ possible permutations, any particular $i$ appears at the $r$-th position exactly $(n-1)!$ times (since there are $(n-1)!$ permutations of the others). Hence, the value corresponding to the uniform strategy $[1/(n!), 1/(n!), \ldots, 1/(n!)]$ for the $r$-th row in $\mathcal{C}$ is

$$
\begin{aligned}
z''_r &= \sum_{c=1}^{n!} \frac{1}{n!} d_{\pi_c[r]} \\
&= \frac{1}{n!} \sum_{c=1}^{n!} d_{\pi_c[r]} \\
&= \frac{1}{n!} \sum_{i=1}^{n} (n-1)! d_{[i]} \\
&= \frac{(n-1)!}{n!} \sum_{i=1}^{n} d_{[i]} \\
&= -\frac{1}{n} S .
\end{aligned}
$$

The third equality applies the above argument of any $i$ appearing at the $r$-th position exactly $(n-1)!$ times. Since this is true for any $r$-th row and that it does not explicitly depend on $r$, the overall value $z'' := \max_r z''_r = -\frac{S}{n}$. Then, the corresponding solution to LP (2) has $z' = -z'' = \frac{1}{n} S$.

For step 3., apply the strong duality (since $z = z' = S/n$) and conclude that the uniform strategies are optimal for both the row and column players. $\qquad\square$

Proposition 3 implies that the uniform strategy is the worst-case optimal approach when selecting a data vendor at the $r$-th position (whose distribution to use as the reference) *without* any prior knowledge or assumptions about the data vendors (i.e., regardless of how the data vendors are ordered).[5]

**Uniform mixture $P_U$ from the uniform strategy.** Since the uniform strategy is worst-case optimal, we implement it via the uniform mixture of the data vendors' distributions $P_U := \frac{1}{n} \sum_i P_i$, as the proposed reference in place of $P^*$ to define Eq. (3) as a practically tractable approximation to $\Upsilon$ (which itself cannot be used since $P^*$ is unknown). In particular, the results on bounded approximation error and actionable policy derived w.r.t. a general $P_\omega$ (i.e., Proposition 2 and Theorem 1) is applicable to $P_\omega = P_U$ since $P_U$ is a special case of $P_\omega$, as the following corollaries.

**Corollary 1.** Let $\hat{\Upsilon} := -d(P, P_U)$, then $\forall P, |\Upsilon(P) - \hat{\Upsilon}(P)| \leq \varepsilon_U d(P^*, Q_U)$.

*Proof of Corollary 1.* The result follows by directly applying Proposition 2 with $\varepsilon_U, Q_U$. $\qquad\square$

**Corollary 2.** Given datasets $D \sim P$ and $D' \sim P'$, let $m := |D|$ and $m' := |D'|$. Let $\hat{\nu}$ be from Eq. (3) where $D_\omega$ is specified as $D_U$: $\hat{\nu}(D) := -\hat{d}(D, D_U)$ and $m_N := |D_U|$. For some bias requirement $\varepsilon_{bias} \geq 0$ and a required decision margin $\varepsilon_\Upsilon \geq 0$, suppose $\hat{\nu}(D) > \hat{\nu}(D') + \Delta'_{\Upsilon, \nu}$ where

---

[5] By the classic [61, Minmax Theorem], these minmax solutions for both players form a Nash equilibrium.

the *criterion margin* $\Delta'_{\Upsilon,\nu} := \varepsilon_\Upsilon + 2[\varepsilon_{\text{bias}} + \sqrt{K/m} + \sqrt{K/m'} + 2\sqrt{K/m_N} + \varepsilon_{\text{U}} d(Q_{\text{U}}, P^*)]$ .
Let $\delta' := 2\exp(\frac{-\varepsilon_{\text{bias}}^2 \overline{m} m_N}{2K(\overline{m}+m_N)})$ where $\overline{m} = \max\{m, m'\}$. Then $\Upsilon(P) > \Upsilon(P') + \varepsilon_\Upsilon$ with probability at least $(1 - 2\delta')$.

*Proof of Corollary 2.* The result follows by directly applying Theorem 1 with $D_{\text{U}}, \varepsilon_{\text{U}}, Q_{\text{U}}$ . □

**Difficulties of searching over all convex mixtures $\omega$.** Note that the game specified in Eq. (4) has a finite action space $N$ for the row player. Essentially, it means that the row player is considering exactly one of the $n$ data vendors as the reference, though the minmax solution suggests that the row player should adopt a mixed strategy to consider all $n$ data vendors equally.

A natural extension of the game is for the row player to consider *all* possible convex mixtures $\omega \in \triangle(n-1)$:

$$\max_{\omega \in \triangle(n-1)} \min_{\pi \in \Pi(N)} -d(P^*, P_{\omega,\pi}) \tag{8}$$

where $P_{\omega,\pi} := \sum_{r=1}^n P_{\pi[r]}\omega_r$ , is a mixture specified by $\omega$ following the permutation $\pi$ , and $\Pi(n)$ denotes the set of all possible permutations of $\{1, 2, \ldots, n\}$ .

From a theoretical perspective (i.e., to understand and analyze the optimal solution, if one exists, to Eq. (8)), there are some difficulties because Eq. (8) is an infinite game (because of the action space of the row player). In particular, infinite games (or semi-finite games where one player has a finite action space) are significantly harder to analyze the optimality or even existence of the (optimal) solution. In general, it is not guaranteed that the optimal solution(s) exist for one or both of the players [62], and in cases where the optimal solutions exist, there may be a so-called duality gap [21]: The minimax theorem for the finite game implies that the optimal solutions for both players will give values that coincide (i.e., the optimal for row player is the negated optimal for the column player), but the duality gap means that in the infinite regime, the optimal solutions may not give values that coincide, making it harder to even verify if a pair of solutions is optimal (since the values may not coincide even if they are indeed optimal).

Moreover, the MMD $d$ in Eq. (8) introduces an additional difficulty since it is not guaranteed to be convex in the mixture of the models: for some $\lambda \in [0, 1], P_1, P_2$, the convex combination of $\lambda d(P^*, P_1) + (1 - \lambda)d(P^*, P_2)$ is not necessarily smaller than or equal to $d(P^*, \lambda P_1 + (1 - \lambda)P_2)$. In other words, there are cases where either direction of the inequality is true. The implication is that, when searching or optimizing over $\omega \in \triangle(n-1)$ via methods that try to "move" a current solution by a small amount to arrive at a new solution (or towards the optimum), it is more difficult to do so, since moving the current solution might increase or decrease the value (and it is not known *a priori* which).

### A.4.3 Additional Discussion on Approximation Error

The result on the approximation error (i.e., Proposition 2) can be useful in designing alternative approaches of finding a reference, and also in considering additional desiderata (e.g., incentive compatibility [5] and truthfulness [14]).

Precisely, for any $P_{\text{ref}}$ (e.g., $P_\omega$ or $P_{\text{U}}$), if it is boundedly close to $P^*$ in the MMD sense, then a bounded approximation error is available (by the triangle inequality of MMD):

$$d(P_{\text{ref}}, P^*) \le \varepsilon_{\text{ref}} \implies \forall P, |\Upsilon(P) - \tilde{\Upsilon}_{\text{ref}}(P)| \le \varepsilon_{\text{ref}} \tag{9}$$

where $\tilde{\Upsilon}_{\text{ref}}(P) := -d(P, P_{\text{ref}})$. We highlight that this precise formalization and the subsequent discussion have not been presented in related existing works (e.g., [1, 14, 60, 66]), and that this discussion is enabled by the analytic properties of Huber and MMD.

**A possible optimization approach for finding a reference and its difficulties.** Intuitively, a smaller approximation error is more desirable (and later we will discuss a specific such desideratum), so naturally we want to minimize it. The following objective is such an example, to be optimized/minimized over all possible convex mixtures of the data vendors' distributions:

$$\min_{\omega \in \triangle(n-1)} \sup_P | -d(P, P^*) - (-d(P, P_\omega))| \tag{10}$$

where $P_\omega := \sum_i \omega_i P_i$ is a convex mixture of the data vendors' distributions as in Observation 1. The triangle inequality of MMD enables a simplification of Eq. (10) to the minimization of an upper bound (of the original objective) instead:

$$\min_{\omega \in \triangle(n-1)} d(P_\omega, P^*) \, .$$

From an implementation or practical perspective, this objective, unfortunately, presents a major practical difficulty in that $P^*$ is *not* available (which is the reason for obtaining a reference in the first place). Hence, additional assumptions are required to make this objective tractable (e.g., a validation dataset from $P^*$ is available [24, 39, 63], each $P_i$ is assumed to be somewhat "close" to $P^*$ [1, 14] or the union or some combination of $\{P_i\}_{i \in N}$ is close to $P^*$ [60, 66]).

We highlight that the assumption $P^*$ is available is indeed a key challenge that we aim to address (by relaxing this assumption) because in practice $P^*$ is *not* available. Hence, exploring a suitable form of the assumption (in the sense that it is practically feasible and also enables a tractable optimization of Eq. (10)) presents an interesting future direction.

**Additional desiderata.** We make precise the intuition that a smaller approximation error is more desirable by connecting the approximation error to the so-called incentive compatibility (IC), frequently used in (truthful) mechanism designs [5, 6, 8, 14]).

**Definition 2** ($\gamma$-incentive compatibility). The valuation function $\Upsilon$ is $\gamma$-incentive compatible, for some $\gamma \geq 0$, if

$$\Upsilon(P_i; \{P_{i'}\}_{i' \in N \setminus \{i\}}, \cdot) \geq \Upsilon(\tilde{P}_i; \{P_{i'}\}_{i' \in N \setminus \{i\}}, \cdot) - \gamma$$

where $\tilde{P}_i$ denotes the mis-reported version of $P_i$.[6]

In Definition 2 [6, 5], $\gamma = 0$ recovers the exact definition of IC. IC is an important desideratum in that it can be used to show that each vendor being truthful (i.e., not misreporting) forms an equilibrium [14] and truthfulness is another such desideratum.

Depending on the specific design, the valuation $\Upsilon$ can have different additional dependencies. For instance, in the ideal case where $P^*$ is available, $\Upsilon(P) := -d(P, P^*)$ has an explicit dependence on $P^*$ and it satisfies $\gamma_{\bar{d}}$-IC where $\bar{d} := \max_i d(P_i, P^*)$ (can be directly verified using Definition 2). On the other hand, for some $\hat{\Upsilon}(P) := -d(P, P_{\mathrm{ref}})$ s.t. $d(P_{\mathrm{ref}}, P^*) = \varepsilon_{\mathrm{ref}}$, it has a *weaker* IC (i.e., a larger corresponding $\gamma$): it satisfies $\gamma_{\mathrm{ref}}$-IC for $\gamma_{\mathrm{ref}} := \varepsilon_{\mathrm{ref}} + \bar{d}$, by the triangle inequality of MMD. Notice that $\varepsilon_{\mathrm{ref}}$ is directly related to the approximation error above, so this suggests that analyzing and then minimizing the approximation error to design a solution with strong IC is a promising future direction. Indeed, some preliminary empirical results (App. D.3.5) demonstrate some promise that (approximate) IC is achievable in some cases.

Note that even in the ideal case where $P^*$ is available, the exact IC is not necessarily guaranteed. This is because the (truthfully reported) distribution $P_i$ by each vendor $i$ is not guaranteed to be the same as $P^*$, which is often the case in practice where the data collected are not guaranteed to be directly from the ground truth distribution [24, 32, 57]. Hence, it is theoretically possible that a "mis-reported" $\tilde{P}_i$ is such that $d(\tilde{P}_i, P^*) < d(P_i, P^*)$, namely an improvement from $P_i$. However, such cases would be rare in practice since it is generally difficult to "move" a distribution $P_i$ (via some statistical operation) towards an *unknown* optimal distribution $P^*$, as otherwise $P^*$ can obtained by simply performing such operations on a given distribution $P_i$ until it reaches $P^*$.

Moreover, [36, 37, 44] have noted that, in our setting of no reference/ground truth (i.e., no $P^*$), it is impossible design a mechanism that guarantees truthfulness of the data vendors, if the data vendors are not assumed to be completely independent of each other. In other words, without access to ground truth, and if there is possibility of so-called side information among the data vendors themselves, a truthful mechanism is impossible, further highlighting the difficulties of our setting (i.e., without a reference).

---

[6] An example of mis-reporting is injecting artificial noise.

# B   Additional Discussion on Related Works

## B.1   Discussion on the Assumptions of Related Works and Applicability to the Problem Statement

In addition to the existing works mentioned in Sec. 2, [7, 31, 32, 72] also share a similar dependence on a given reference dataset. Consequently, it is unclear how to apply these methods *without* a given reference dataset.

Next, we elaborate the works that relax the assumption of a given reference and better contrast their differences from our work, specifically in how they relax this assumption.

Chen et al. [14, Assumption 3.2] require that $P^*$ must follow a known parametric form and the posterior of the parameters is known. Chen et al. [14] assumes that each data vendor collects data from the ground truth data distribution (parametrized by some unknown parameters) and performs the analysis on whether a data vendor $i$ will report untruthfully when *everyone* else is reporting truthfully. Specifically, the reference is the aggregate of all vendors excluding the vendor $i$ itself. It is unclear how heterogeneity can be formalized in their theoretical analysis, which assumes the vendors are collecting data from the same (ground truth) data distribution. [60] directly assumes that the aggregate dataset $D_\omega$ or the aggregate distribution $P_\omega$ is a sufficiently good representation of $P^*$ and thus provides a good reference, *without* formally justifying it or accounting for the cause or effect of heterogeneity. Wei et al. [66, Assumption 3.1] require that for any $i, i'$, the densities of $P_i$ and $P_{i'}$ must lie on the same support, which is difficult to guarantee because $Q_i, Q_{i'}$ can have different supports (resulting in $P_i, P_{i'}$ having different supports). [66] uses the $f$-divergence specifically the KL divergence, which requires an assumption that the densities of the data distributions (of all the vendors) to lie on the same support [66, Assumption 3.1]. This assumption can be difficult to satisfy in practice since the analytic expression of the density of the data distribution is often complex and unknown. To illustrate, our experiments consider heterogeneity in the form of mixture between MNIST and FaMNIST; it is unclear how to satisfy the assumption that the densities of the distributions of MNIST and FAMNIST lie on the same support. Moreover, their method does not formally model heterogeneity or its effect on the value of data.

Furthermore, a similarity in these works [14, 60, 66] is how they leverage the "majority" to construct a reference either implicitly or explicitly. Specifically, in [14], the reference for vendor $i$ is the aggregate of every vendor's dataset except $i$; in [60, 66] the reference is constructed by utilizing both the aggregate dataset $D_\omega$ and additionally some synthetically generated data (Tay et al. [60] use the MMD-GAN while Wei et al. [66] use the $f$-divergence GAN). We highlight that these works did not provide a theoretical analysis on "how good" their respective reference is, namely, what is the error from using the reference instead of using the ground truth? In contrast, we answer this question via Proposition 2 and propose to use the uniformly weighted "majority" as the solution, inspired by the worst-case optimality of the uniform strategy in the two-player zero-sum game (Proposition 3).

Note that there are recent works that emphasize different aspects such as privacy [58] and without access to data [71], and hence differ from our settings.

## B.2   Comparison with Alternative Distances/Divergences

Supplementing the comparison of Table 1, we provide additional details and discussion with alternative distances/divergences (to our adopted MMD) that have already been adopted for the purpose of data valuation.

**Comparison between Ours and MMD$^2$.**   Empirically, ours (i.e., MMD-based) and MMD$^2$ [60] perform similarly across the investigated settings (including empirical convergence and ranking data distributions), which is unsurprising since theoretically the numerical values only differ by a square. Although MMD$^2$ has an unbiased estimator [26, Eq. (4)] while MMD, to our knowledge, only has a biased estimator [26, Eq. (6)], this advantage does not seem significant, since the convergence results in Sec. 5.1 demonstrate similar convergences for Ours and MMD$^2$. Recall that Sec. 5, we highlighted that the implemented estimator for MMD$^2$ is *not* obtained from taking the square of that for MMD. Nevertheless, we have also tried this implementation of directly squaring the estimator for MMD to be the estimator of MMD$^2$ but did not observe a significant difference in the empirical results.

On the other hand, from a theoretical perspective, the difference in terms of the implications, is more significant. This is primarily due to the analytic properties of MMD, which are *not* also satisfied by $MMD^2$, such as the triangle inequality, used to derive Proposition 2, and the property with Huber model (i.e., Eq. (2)) which is used to derive Lemma 4 and Theorem 1. It is an interesting future direction to explore similar results for $MMD^2$.

**Comparison between Ours and the Wasserstein metric.** Similar to MMD, the Wasserstein metric, also known as the optimal transport (OT) distance [34] also satisfies the axioms of a metric, in particular the triangle inequality. This can make the Wasserstein metric a promising choice for our setting. However, MMD seems to have two important advantages —one theoretical and the other practical—: (i) Under the Huber model, MMD enables a simple and direct relationship (i.e., Eq. (2)) that precisely characterizes how the heterogeneity (formalized by Huber) of a distribution affects its value. It is unclear how or whether the Wasserstein metric can provide the same relationship, and some works [50, 51] seem to suggest the difficulties of obtaining the same relationship. (ii) The definition of the Wasserstein metric (involving taking an infimum over the couplings of distributions) makes its value difficult to obtain (compute or approximate) in practice. Indeed, LAVA's official implementation [33, Github repo] does not directly compute/approximate the 2-Wasserstein distance, but instead obtains the calibrated gradients as a surrogate [33, Sec. 3.2], which are not explicitly guaranteed to approximate the 2-Wasserstein-based values. In contrast, our method directly approximates the MMD (with the estimator [26, Eq. (6)]) with a clear theoretical guarantee (Lemma 1). In other words, though the Wasserstein metric satisfies the appealing theoretical properties, in implementation, a valuation based on the Wasserstein metric is difficult to obtain directly, and instead some surrogate is obtained. Moreover, it has not yet been guaranteed that this surrogate also satisfies (possibly approximately) the same appealing theoretical properties that might make the Wasserstein metric a promising choice.

**Comparison between Ours and $f$-divergences.** The $f$-divergences family presents a rich choice (since it contains many specific divergences such as Kullback-Leibler, total variation and etc.) and is also adopted in existing works such as [14, Definition 4.5], [66, Algorithm 1] and [1, Eq.(1)] for the theoretical properties of the adopted $f$-divergence (or variant). For instance, the Kullback-Leibler (KL) divergence satisfies the required [66, Assumption 3.4] for the proposed method, while the extended KL divergence proposed by Agussurja et al. [1, Sec. 3] enables a decomposition of the terms to simplify the analysis.

These theoretical properties notwithstanding, our adopted MMD has a clear advantage over the $f$-divergence with important practical implications. A commonly made assumption with using the $f$-divergence $f(P\|Q)$ is the absolute continuity of $P$ w.r.t. $Q$, since otherwise the division-by-zero makes the definition ill-behaved. This assumption is difficult to satisfy or even verify in practice, especially for complex and high-dimensional data distributions. In contrast, the sample-based definition of MMD does *not* require such an assumption, making its application to complex and high-dimensional data distributions easier (in the sense that the user does not have to worry about a difficult-to-satisfy assumption). Intuitively, this difference between $f$-divergences and most integral probability metrics (IPMs) such as MMD, is that when $P$ and $Q$ have disjoint supports, all $f$-divergences take on a constant value; in contrast, IPMs can give "partial credit".

Another important practical implication due to the difference in the definitions of $f$-divergences and MMD is that: it is more direct and easier to approximate MMD (e.g., using a sample-based approach [26, Eq. (6)] and with a theoretical guarantee as in Lemma 1). In contrast, the definition of the $f$-divergence $f(P\|Q)$ that directly depends on the density functions of $P, Q$ adds to the difficulties of estimating it in practice, as it requires estimating the density functions (or at least the ratio of the density functions). This is difficult for complex and high-dimensional data distributions and may require simplifying assumptions of the parametric form of the distributions (e.g., $P, Q$ are both multi-variate Gaussian [1] to enable a closed-form expression for KL).

## B.3 Comparison with other Characterization/Treatment of Data Heterogeneity

As an additional comparison to supplement our remarks in Sec. 3 for why adopting the Huber model, we highlight a comparison with relevant existing methods in their treatment of data heterogeneity, and then further elaborate on the analysis based on the design choice of the Huber model.

Recall that the Huber model characterizes a sampling distribution $P$ via a mixture (weighted by $\varepsilon$) between the ground truth distribution $P^*$ and some unknown distribution $Q$ (e.g., an outlier distribution). This characterization is sufficiently general to model several sources of heterogeneity (via $\varepsilon$ and $Q$) [12, 13], while also having a dependence on the ground truth distribution $P^*$ of interest. In our setting, this modeling choice means that the sampling distributions of the different data vendors have varying "qualities" via their different dependence on the ground truth distribution $P^*$ (in pathological cases where $\varepsilon \to 1$, $P$ effectively has no dependence on the ground truth distribution and only contains $Q$). This way, Huber model provides a precise yet relatively general way to characterize the differences, namely heterogeneity in the sampling distributions of the data vendors: their sampling distributions may have different dependence on $P^*$, and this dependence is made precise via $\varepsilon, Q$. This precise characterization is important in enabling the subsequent analysis.

**Existing works do not precisely characterize data heterogeneity.**    Before highlighting the analysis based on the Huber model and the theoretical appeal thereof, we first compare with some relevant existing works, which either do *not* characterize the heterogeneity at all [2, 33, 55, 66, 67] , or do so through simplifying assumptions [1, 14, 60].

Agussurja et al. [1, Assumption (**A3**)] assume that in the infinite sample regime, for the data from each data vendor, there exists a so-called uniformly consistent distinguisher. In other words, it means, for the purpose of parametric estimation (which is the setting in [1]), the sampling distribution of each vendor has the same quality, this is because with sufficient samples, each vendor individually can identify or recover the true parameters. In summary, this assumption simplifies the problem and bypasses heterogeneity by making the sampling distribution of each data vendor the same specifically w.r.t. parametric estimation (i.e., these sampling distributions are not the same in general).

Chen et al. [14, Section 3, Assumption 3.1] assume that the dataset of each vendor consists of i.i.d. samples conditioned on some common unknown parameters $\boldsymbol{\theta}$. The authors also note that "This (Assumption 3.1) is definitely not an assumption that would hold for arbitrarily picked parameters $\boldsymbol{\theta}$ and any datasets." In this way, it is unclear how the heterogeneity of the sampling distribution of each individual data vendor is precisely characterized.

While Tay et al. [60] consider the sampling distributions of the vendors to be heterogeneous, the authors do not precisely define the heterogeneity. As the treatment, Tay et al. [60, Assumption (B)] assume that the aggregate distribution (i.e., the underlying distribution of the union of the samples from all the data vendors) "approximates the true data distribution well." Note that the authors do not precisely define what it means by approximating the true data distribution well, in contrast we provide such an analysis (i.e., Proposition 2 for the general class of convex mixtures).

**The (appeal of the) analysis based on the Huber model.**    We elaborate further on the property (e.g., Lemma 4) and the results (e.g., Eq. (2)) based on the Huber model.

In our studied setting with multiple data vendors, the mixture model of Huber is particularly useful because a mixture of Huber distributions is also a Huber, whose parameters (i.e., $\varepsilon, Q$) can be derived based on the parameters of the component Huber distributions, namely via Lemma 4. This makes Huber particularly appealing for the purpose of theoretical analysis. To elaborate, given a data valuation function (e.g., Eq. (1)) that is well suited for Huber distributions, we can use this to evaluate the value of the distribution of a single data vendor, or any specified convex mixture of distributions of multiple data vendors (which is useful for the peer-prediction paradigm [14, 60, 66] and adopted in Sec. 4).

Subsequently, our designed MMD-based valuation function in Eq. (1) exploits the structural property of the Huber model (i.e., it is a mixture) to provide a precise characterization of the effect of such heterogeneity on the value of data, namely Eq. (2). In other words, Eq. (2) answers the question: "precisely how does the heterogeneity of a data distribution affect its value?" This has *not* been achieved in prior works.

Additionally, the theoretical properties the Huber model (jointly with the MMD-based valuation) admit the results that describe an actionable policy for comparing different sampling distributions (i.e., Proposition 1, Theorem 1), and also a game-theoretic perspective that yield the uniform mixture as the worst-case optimal Proposition 3. We highlight that such results leverage certain specific mathematical properties such as Lemma 4 and Eq. (2).

**Connection to robust statistics and a possible error lower bound.** We highlight that mixture models have deep roots in the field of robust statistics [19, 20, 40] where the goal is to estimate some statistics (e.g., mean and covariance) of the ground truth distribution $P^*$. Hence, prior results obtained can shed light on our setting, method and results. For instance, [40] prove an information-theoretic lower bound of error for mean estimation using the data collected from the mixture model (between $P^*$ and some outlier distribution $Q$). In particular, for a sampling distribution $P := (1 - \varepsilon)P^* + \varepsilon Q$, the error (of mean estimation) has a lower bound that is at least linear in the proportion $\varepsilon$ of the outlier distribution [40, Observation I.4]. For our setting, this result can suggest that it is impossible to obtain an arbitrarily good reference $P_\omega$ s.t. $d(P_\omega, P^*) < \varepsilon_{\mathrm{MMD}}$ for any desired $\varepsilon_{\mathrm{MMD}} \geq 0$ where the mixture model is as in Observation 1. This can further justify our approach of using a convex mixture $P_\omega$ or $P_{\mathrm{U}}$ as the reference, which would have an "error" (i.e., $d(P_\omega, P^*)$) linear in $\varepsilon_\omega$ (i.e., Lemma 4 or Corollary 1). Note that here a main difference is that we have multiple data vendors. We need to consider a collection of distributions $\{P_i\}_{i=1}^n$ and outlier distributions $\{Q_i\}_{i=1}^n$ where in contrast, [20, 40] consider settings with only one outlier distribution $Q$. This means that we need to additionally consider how to "combine" these distributions, which we do so via their convex mixtures as in Observation 1).

We outline an informal sketch of this idea and defer the formal treatment to future work, which requires a careful "translation" between the settings of mean estimation and distribution estimation, see below: The results in [20, 40] are w.r.t. the $\ell_2$-norm of the error for mean estimation. In our setting, the error (i.e., the MMD $d(P_\omega, P^*)$) is w.r.t. the RKHS norm (MMD is an RKHS norm [26, Lemma 4]). Note that, for a function $f \in \mathcal{H}$ in an RKHS $\mathcal{H}$, its $\ell_2$-norm is dominated by its RKHS norm: $\|f\|_{\ell_2} \leq \|f\|_{\mathcal{H}}$ [17, Section 2]. In other words, one can transfer the information theoretic lower bound (e.g., [40, Observation I.4]) to our setting by using the fact that the MMD (which is an RKHS norm) dominates the $\ell_2$-norm of the error of mean estimation. Informally, since the $\ell_2$ error of mean estimation is lower bounded, and that the RKHS norm is larger than (or equal to) the $\ell_2$-norm, the RKHS norm (i.e., the MMD between the any reference $P_\omega$ and $P^*$) is also lower bounded, meaning that it is impossible to find an arbitrarily good reference $P_\omega$.

## C  Questions & Answers

Due to page constraints of the main paper, we provide some additional elaboration to supplement our main paper and without which would not affect the completeness of the treatment in our main paper. We adopt the form of questions and answers for the convenience of the reader.

**Q1.** Why is there a need for explicitly considering a valuation method for a data distribution? Why not directly extend existing dataset valuation methods to data distribution valuations?

**Answer**: In addition to the practical use-cases outlined in Sec. 1 which require a value of a data distribution instead of a value for a finite dataset, there are theoretical considerations for the perspective of data distribution valuation, as elaborated next.

Most existing works focus on defining a value $\nu(D)$ for a fixed discrete dataset $D$ instead of the value $\Upsilon(P)$ for the sampling distribution $P$ from which each individual data point in $D$ is i.i.d. sampled. Then, one might think to extend a defined/given dataset valuation $\nu$ to define a value $\Upsilon(P)$ of $P$, such as based on the expectation of $\nu(D)$ over the randomness of $D$: For some existing/already defined $\nu(D) \mapsto \mathbb{R}$,

$$\Upsilon(P) := \mathbb{E}_{D \sim P}\nu(D) . \tag{11}$$

Additionally, to account for the fact that $D$ has a finite size, consider the limit of this expectation as the size $|D|$ of $D$ approaches infinity, since in the infinite sample regime, the sample statistic obtained on $D$ would tend to the population statistic on $P$:

$$\Upsilon(P) := \lim_{|D| \to \infty} \mathbb{E}_{D \sim P}\nu(D) . \tag{12}$$

Such definitions in Eq. (11) and Eq. (12) require a theoretically defined $\nu$ to understand and study the theoretical properties of the corresponding $\Upsilon(P)$, which is often not available, and also a characterization of $P$ due to the $D \sim P$ in the expectation, which is also not available.

To elaborate, the most commonly adopted choices of $\nu$, following [24], are defined through some empirical observations, e.g., the evaluation performance on a given validation set $D_{\mathrm{val}}$ of a fixed ML

model trained on $D$. While for the purpose of implementation and experiments, for thusly defined $\nu$, Eq. (11) can be approximated reasonably well, there are difficulties when it comes to obtaining a theoretical analysis of $\Upsilon$, primarily because the analytic expression of $\nu$ is difficult to obtain. This is because this analytic expression of $\nu$ depends on the choice of $D_{\text{val}}$, the ML model, the training procedure and etc. and can be complicated. An existing work proposes to utilize the framework of neural tangent kernel to help characterize $\nu$ [67], but nevertheless does not consider or define $\Upsilon$ or explicitly characterize the sampling distribution $P$.

Secondly, Eq. (11) and Eq. (12) require a characterization of $P$ (because of $\mathbb{E}_{D \sim P}$), which has not been proposed or studied by prior works.

Therefore, instead of trying to extend the existing valuation methods defined for discrete datasets to data distributions, which would encounter the difficulties (of obtaining the analytic expression of $\nu$), we adopt the perspective of considering the distributions directly and propose a valuation for data distributions. The advantage of this perspective is that, a valuation defined for a distribution automatically yields a valuation for a dataset because the value of a distribution is essentially a population statistic while the corresponding value for the dataset is the sample statistic and is well defined. In other words, we propose the perspective of "going from data distributions to datasets", because of the appeal for theoretical analysis.

**Q2.** What is the right interpretation of the problem statement and how it is connected to the theoretical results of Proposition 1 and Theorem 1?

The problem statement asks for a *prescriptive* solution: A result that informs the buyer, in order to make the determination that $P$ is more valuable than $P'$ by some amount $\varepsilon$ (i.e., this is what the buyer wants), here is what the buyer needs to observe. This is made precise by Proposition 1 and Theorem 1 where the amount $\varepsilon$ is denoted as $\varepsilon_{\Upsilon}$, and what the buyer needs to observe is that $\nu(D) > \nu(D') + \Delta_{\Upsilon, \nu}$. Suppose that the buyer observes that this inequality is not satisfied, then the buyer could decide not to make the decision, make the decision but with a lower confidence (i.e., larger $\delta$), or request that the vendors to provide a larger sample size, as elaborated in the paragraphs following Proposition 1 and Theorem 1 respectively.

We note that our results, by design, differ from an *inferential* policy where a buyer, upon passively observing $\nu(D), \nu(D')$ and their difference $\nu(D) - \nu(D')$, tries to infer the difference $\Upsilon(P) - \Upsilon(P')$.

We believe that in our motivated settings (i.e., data marketplaces), a prescriptive policy is more useful, since the buyer can use it to make proactive decisions/actions (e.g., specifying how much more valuable $P$ than $P'$ to purchase $P$, requesting a larger sample from the vendor), instead of being a passive observer (such as in the inferential policy).

**Q3.** Why is there a hypothetical column player?

**Answer**: The hypothetical column player is used to clearly represent the fact that the buyer (i.e., the row player) does *not* have prior knowledge about the vendors. Effectively, to the buyer, before having seen or bought the vendors' distributions, all vendors are equivalent. Then, in this two-player game formulation, the column player (via its action space of all possible permutations) effective models this fact. The implication is that this finite, two-player zero-sum game can be analyzed *without* explicitly knowing the payoff matrix $\mathcal{R}$ to show that the uniform strategy is indeed worst-case optimal (Proposition 3).

**Q4.** Can some prior knowledge about the data vendors be used to design a better solution than $P_{\text{U}}$?

**Answer**: In principle, yes. However, it is yet unclear precisely what form of prior knowledge to consider, and then *how* to exploit it to design a better solution. One possibility to build on top of the two-player game formulation is to apply a prior belief over the rankings of the vendors in terms of the qualities of their distributions and then try to solve the corresponding game. It thus forms an interesting future direction to explore.

**Q5.** Is the uniform mixture optimal?

**Answer**: The uniform strategy is worst-case optimal in the game defined as in Eq. (4), and it inspired our choice of the uniform mixture. A possible extension is to the infinite game version where the row player searches for some "optimal" $\omega$ over $\triangle(n-1)$ (formalized via Eq. (8) and elaborated in

App. A), but it is faced with additional difficulties, as detailed in the discussion following Eq. (8). It is not yet clear whether the uniform mixture would be (a part of) the optimal solution to the more complex infinite game, and thus presents an interesting exploration.

**Q6.** Given the setting of no ground truth, and no prior knowledge about the data vendors, intuitively it does seem that one cannot do better than the uniform strategy. Then, is there some lower bound (of error) that matches the upper bound of error (e.g., Proposition 2 or Corollary 1)?

**Answer:** We provide an informal discussion based on two different perspectives (one from the field of robust statistics and the other from mechanism design).

Firstly, in the field of robust statistics, a lower bound of error for some estimation problem is often studied and there is indeed literature specifically studying the Huber model. But the differences in settings make their results not directly applicable, though instructive for us. For instance, [40] derive an information-theoretic lower bound of error for mean estimation using data sampled from a Huber model $P := (1 - \varepsilon)P^* + \varepsilon Q$ where the lower bound is at least linear in $\varepsilon$ [40, Observation I.4], but does not have a result on the statistical difference between $P^*$ and $Q$. Nevertheless, this lower bound seems to already match our upper bound (e.g. Proposition 2) w.r.t. $\varepsilon$ (i.e., linear). Two key differences in the setting are: (i) we have multiple data vendors where [40] study only one, so we need to additionally consider how to "combine" these data vendors; (ii) the result in [40] is w.r.t. the $\ell_2$-norm, which is different from MMD (an RKHS norm) though there are similarities that can be used. We expand on how these differences make it difficult to directly apply the results of [20, 40], but also outline an informal sketch of how to adapt their results in App. B.3, which could be an interesting future direction.

Secondly, in the field of mechanism design where some interaction among multiple "agents" (i.e., data vendors in our setting) is typically present, a similar result (though in a different flavor from the error bound in robust statistics) is also observed. [36, 37, 44] point out that it is indeed impossible for a mechanism (e.g., a designed data valuation) to guarantee truthfulness if (i) there is no ground truth, and (ii) the data vendors are not assumed to be independent. Here truthfulness is related to the error of a valuation function where if the valuation function has $0$ error (i.e., we manage to find some $P_\omega$ s.t. $P_\omega = P^*$), truthfulness is guaranteed. By the contra-positive of this impossibility, without making further assumptions (the precise form of which is yet unclear), in our setting, it is impossible to design a valuation function that has an arbitrarily low error, or equivalently to find some $P_\omega$ with an arbitrarily small MMD to $P^*$. While this result is not quantitative as the error lower bound from robust statistics, it is useful nevertheless because it applies to our setting with multiple data vendors, no prior knowledge about the data vendors and no ground truth.

We hope that our discussion here and our work can generate more interest in exploring these interesting directions (e.g., to obtain a matching lower bound of error), and that our described framework and proposed method can provide a theoretical basis for such future explorations.

**Q7.** Instead of Eq. (4), why can't an optimization/search be performed directly over the convex mixture of $P_i$'s by assuming that $\exists \omega^* \in \triangle(n-1)$ s.t. $P_{\omega*} = P^*$?

**Answer:** Suppose that $\forall i, \varepsilon_i \neq 0, Q_i \neq P^*$, then it can be shown that any convex mixture of $P_i$'s is *not* $P^*$, because any convex mixture necessarily contains a (mixture) component that is *not* $P^*$ (i.e., $\varepsilon_\omega Q_\omega$ in Observation 1). Now, suppose it is relaxed that for some $i^*$, $P_{i^*} = P^*$ . Then the main problem reduces to finding that $i^*$ out of $n$ vendors *without* having prior knowledge about the vendors, which indeed corresponds to the game described by Eq. (4). Similarly, the same game can be generalized to to finding $\arg\min_i d(P^*, P_i)$ even if $\min_i d(p_i, P^*) \neq 0$ . Essentially, in both cases (either $\min_i d(P_i, P^*) = 0$ or $\min_i d(P_i, P^*) \neq 0$), the same described by Eq. (4) can be instantiated for which Proposition 3 is applicable.

Regarding the direct search over $\exists \omega^* \in \triangle(n-1)$, an additional discussion that suggests some additional assumptions (with yet unclear precise forms) may be necessary, is included in App. A.4.3.

**Q8.** (How) can this method be applied if the sample datasets of the vendors have different sizes?

**Answer**: Yes: Suppose that the sample datasets $D_i$'s have different sizes, then it is possible to use the minimum size $m_{\min}$ of these sample datasets and uniformly randomly select a subset $D_{i,\text{sub}}$ from each $D_i$ of size $m_{\min}$ so that the resulting subsets $D_{i,\text{sub}}$'s have equal sizes. This is mentioned as an

implementation detail in Sec. 5 and hence in our experiments, w.l.o.g. we assume that the sample datasets have the same size.

**Q9.** Is this method still effective if the Huber model is *not* satisfied?

**Answer**: While our theoretical results are specific to the Huber model (to exploit the analytic properties of MMD and Huber, e.g., in Proposition 2), some preliminary empirical results (in App. D) under two specific non-Huber settings do demonstrate that our method can remain effective even if the Huber model is not satisfied.

**Q10.** What are the difficulties of extending the theoretical results beyond the Huber model?

**Answer**: There are two main difficulties: (1) how to provide a formal treatment of the interactions of multiple vendors? with the Huber model, Observation 1 suggests that the convex mixture is one effective way. (2) how to analyze the effect of heterogeneity on the value of data? with the Huber model, Eq. (2) provides a simple yet intuitive way to do so.

**Q11.** How is the extension "Ours cond." different?

**Answer:** Our method (denoted as Ours) focuses on the features of the sample datasets and does *not* exploit the label information in these datasets. In contrast, some baselines such as (LAVA and CS) do exploit the label information. The extension (denoted as Ours cond.) is to demonstrate that our proposed MMD-based approach can be extended to also exploit the label information, if available. In essence, while "Ours" uses the MMD between the feature distributions, "Ours cond." uses the MMD between the conditional distributions (i.e., distribution of label conditioned on feature). The implementation details are provided in Sec. 5 and App. D.

**Q12.** What is the difference between $D_{\text{test}}$ and $D_{\text{val}}$?

**Answer**: $D_{\text{val}}$ is included to accommodate some baselines that require it in order to enable a comparison. We highlight that it is an assumption that these baselines require in their works, and our method does *not* require $D_{\text{val}}$. In contrast, $D_{\text{test}}$ is used *only* for the purposes of evaluation so that we can compare different baselines (i.e., valuation methods). $D_{\text{test}}$ is *not* required to implement our method.

**Q13.** Why is there a need to use a common ground truth $\zeta$ and what is the interpretation of it being the expected test performance?

**Answer**: The purpose of $\zeta$ is for the evaluation and comparison of multiple methods as in Sec. 5. In practice, none of the methods can have a dependence on $\zeta$.

For its purpose of evaluation and comparison, $\zeta$ is meant to denote a sensible and interpretable value (i.e., ground truth) for the distributions. A sensible value refers to that the ground-truth value of $\zeta_i$ for $P_i$ should make sense. For instance, a data distribution $P_i$ with a high $\varepsilon_i$ and a high $d(P^*, Q_i)$ should not have a high ground-truth value $\zeta_i$. An interpretable value requires $\zeta$ to be simple and directly understandable since $\zeta$ will be used as a reference, and if it is not interpretable, then it is difficult to draw analysis of the comparison obtained using $\zeta$ as the reference.

As such, in our empirical investigation, we implement $\zeta$ as the expected test performance which is interpretable since test performance directly informs the user how well the model trained on some data performs, and it is sensible as it is adopted by existing methods [25, 57, 72].

# D  Additional Experimental Settings and Results

## D.1  Dataset Licenses and Computational Resources

MNIST [41]: Creative Commons Attribution-Share Alike 3.0. EMNIST [16]: CC0: Public Domain. FaMNIST [68]: The MIT License (MIT). CIFAR-10 and CIFAR-100 [38]: The MIT License (MIT). CaliH [35]: CC0: Public Domain. KingH [28]: CC0: Public Domain. TON [47]: CC0: Public Domain. UGR16 [45]: CC0: Public Domain. Credit7 [49]: CC0: Public Domain. Credit31 [3]: CC0: Public Domain. Census15, Census17 [48]: CC0: Public Domain.

Our experiments are run on a server with Intel(R) Xeon(R) Gold 6226R CPU @2.90GHz and 4 NVIDIA GeForce RTX 3080's (each with 10 GBs memory). We run our experiments for 5 independent trials to report the average and standard errors.

## D.2 Additional Experimental Settings

Table 4 provides an overall summary of the experimental settings.

Table 4: Datasets, used ML models $\mathbf{M}$ and $n, m_i, \varepsilon_i$.

| Setting | $P^*$ | $Q$ | $\mathbf{M}$ | n | $m_i$ | $\varepsilon_i$ |
|---|---|---|---|---|---|---|
| Class. | MNIST | EMNIST | CNN | 5 | 10000 | $(i-1)/n$ |
| | MNIST | FaMNIST | CNN | 10 | 10000 | $(i-1)/n$ |
| | CIFAR10 | CIFAR100 | ResNet-18 | 5 | 10000 | $(i-1)/n$ |
| | Credit7 | Credit31 | LogReg | 5 | 5000 | $(i-1)/(4n)$ |
| | TON | UGR16 | LogReg | 5 | 4000 | $(i-1)/(4n)$ |
| Regress. | CaliH | KingH | LR | 10 | 2000 | $(i-1)/n$ |
| | Census15 | Census17 | LR | 5 | 4000 | $(i-1)/n$ |

**Additional dataset preprocessing details.** For CaliH and KingH, the pre-processing is passing the original (raw) data through a neural network (i.e., feature extractor) with the last layer having 10 units [1, 70]. Hence, the dimensionality after preprocessing is 10.

Credit7 has 7 features while Credit31 has 31 features, so the top 7 principal components are kept for Credit31. TON and UGR16 have the same feature space including features such as packet byte, source and destination IP addresses, but non-numerical features such as IP addresses are converted to one-hot encoding.

For TON and UGR16, which share the same feature space containing features such as source and destination IP addresses, packet size, network protocol and etc, we adopt the one-hot encoding for non-numerical features (i.e., source and destination IPs, network protocol) and perform standard scaling of the numerical features (i.e., packet size). The dimensionality after preprocessing is 22.

**ML model $\mathbf{M}$ specification details.** For MNIST vs. EMNIST and MNIST vs. FaMNIST, a standard 2-layer convoluntional neural network (CNN) is adopted. For CIFAR10 vs. CIFAR100, the ResNet-18 [29] is adopted. For both Credit7 vs. Credit31, and TON vs. UGR16, a standard logistic regression with the corresponding input sizes is adopted. The specific implementation (i.e., source code) is provided in the supplementary material for reference.

**Additional implementation details on Ours cond.** Different from directly computing MMD between the features, this extension of our method aims to additionally utilize the label information contained within each $D_i$. Recall that each single data point in $D_i$ is paired feature-label, so this extension aims to exploit such information. Theoretically, MMD is well-defined w.r.t. distributions, be it distributions over only the features (i.e., $P_X$) or conditional distributions of the labels given features (i.e., $P_{Y|X}$). For implementation, we need a representation for $P_{Y|X}$ for a $D_i$. In our implementation, we train a machine learner $\mathbf{M}_i := \mathbf{M}(D_i)$ on $D_i$ and use it to construct the empirical representation for $P_{Y|X}$. Given a reference $D_{\text{val}}$, we collect the set of predictions (denoted as $\mathbf{M}_i(D_{\text{val}})$) of $\mathbf{M}_i$ on this reference, as the empirical representation of $P_{Y|X}$. Subsequently, we compute the data values, namely negated MMD between $\mathbf{M}_i(D_{\text{val}})$ and, the labels of $D_{\text{val}}$ (i.e., the columns under $\rho(\boldsymbol{\nu}, \boldsymbol{\zeta})$ in Table 2). Note that the predictions (i.e., $\mathbf{M}_i(D_{\text{val}})$) are probability vectors in the $C-1$-probability simplex $\triangle(C-1)$ for classification with $C$ classes, or real-values for regression. For MMD computation, a one-hot encoding of the labels in $D_{\text{val}}$ for classification is performed; no additional processing is required for the labels for regression.

**Reproducibility statement.** We have included the necessary details to ensure the reproducibility of our theoretical and empirical results. Regarding theoretical results, the full set of assumptions, derivations and proofs for each theoretical result is clearly stated in either the main paper, or App. A.

Regarding experiments: (i) the code to produce the experiments is included in a zip file as part of the supplementary material. It also contains the code and scripts to process the data used in the experiments. (ii) the processing steps and the licenses of the datasets used in the experiments, and the parameters (e.g., the choice of ML model used) that describe our experimental settings are clearly described in App. D. (iii) The information of the computational resources (i.e., hardware) used in our experiments and a set of scalability results for our method are included in App. D.

### D.3 Additional Experimental Results

#### D.3.1 Additional Results for Empirical Convergence

Figs. 3 to 5 demonstrate that our method (i.e., MMD) performs overall the best (converges the most quickly) on TON, CIFAR10 and Credit7, respectively. Note that on these three datasets (i.e., CIFAR10, TON and Credit), the baseline LAVA did not complete due to a known runtime error (see Github issue) of a required package OTDD.

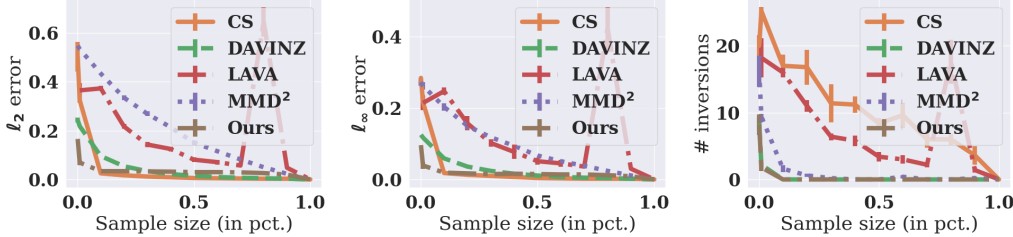

Figure 2: The 3 criteria (on $y$-axis) for $P^* =$ MNIST vs. $Q =$ FaMNIST. $n = 10, m_i^* = 10,000$. $x$-axis shows sample size in percentage, i.e., $m_i/m_i^*$ where $m_i^*$ is fixed to investigate how the criteria change w.r.t. $m_i/m_i^*$: If the criteria decrease quickly w.r.t. $m_i/m_i^*$, it means the metric converges quickly (i.e., sample-efficient). Averaged over 5 independent trials and the error bars reflect the standard errors.

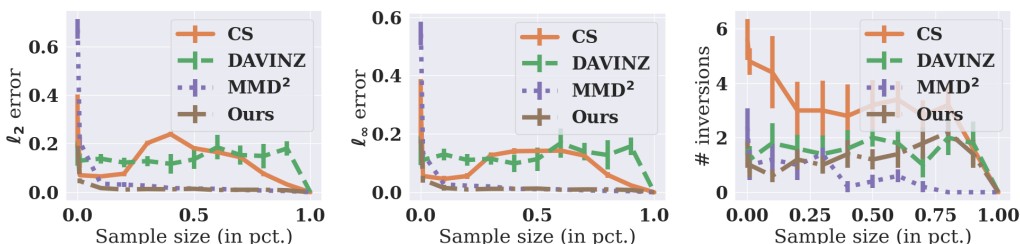

Figure 3: The 3 criteria (on $y$-axis) for $P^* =$ TON vs. $Q =$ UGR16. $n = 5, m_i^* = 10,000$. $x$-axis shows sample size in percentage, i.e., $m_i/m_i^*$ where $m_i^*$ is fixed to investigate how the criteria change w.r.t. $m_i/m_i^*$: If the criteria decrease quickly w.r.t. $m_i/m_i^*$, it means the metric converges quickly (i.e., sample-efficient). Averaged over 5 independent trials and the error bars reflect the standard errors.

#### D.3.2 Experimental Verification of Correlation between True Values vs. Distribution Errors

This experiment serves as a preliminary verification on the empirical suitability of our proposed valuation in Eq. (1): Whether/how well do the measured values (i.e., $\hat{\nu}$ in Eq. (3)) correlate with the error levels (i.e., $d(P, P^*)$)? The specific settings are as follows,

1. We consider discrete distributions for $P^*, P_i, Q_i$. We randomly generate 1-dimensional discrete (not necessarily uniform) distributions supported on the integers in $[0, 10]$. We first generate $P^*$. Then, for each vendor $i$, we randomly generate another distribution as $Q_i$ and also an $\varepsilon_i \sim \text{Uniform}(0, 0.5)$ (so that the outlier is not overpowering the ground truth) for $P_i = (1 - \varepsilon_i)P^* + \varepsilon_i Q_i$.

2. The error level of distribution is measured by $d(P_i, P^*)$.

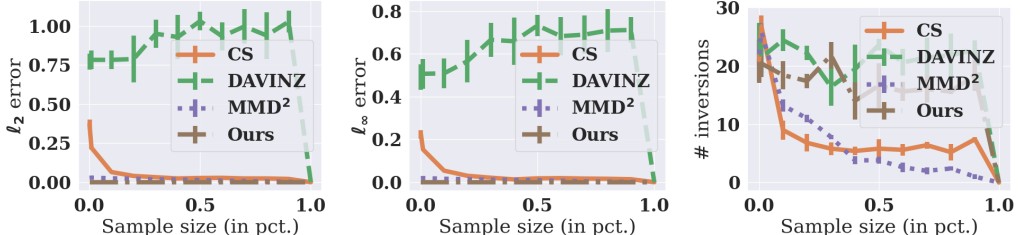

Figure 4: The 3 criteria (on $y$-axis) for $P^* = $ CIFAR10 vs. $Q = $ CIFAR100. $n = 10, m_i^* = 10,000$. $x$-axis shows sample size in percentage, i.e., $m_i/m_i^*$ where $m_i^*$ is fixed to investigate how the criteria change w.r.t. $m_i/m_i^*$: If the criteria decrease quickly w.r.t. $m_i/m_i^*$, it means the metric converges quickly (i.e., sample-efficient). Averaged over 5 independent trials and the error bars reflect the standard errors.

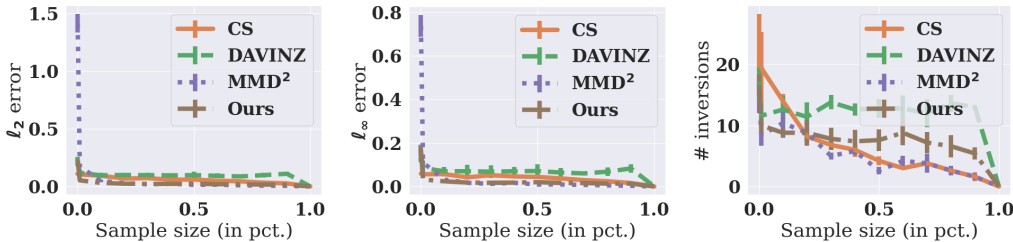

Figure 5: The 3 criteria (on $y$-axis) for $P^* = $ Credit7 vs. $Q = $ Credit31. $n = 10, m_i^* = 10,000$. $x$-axis shows sample size in percentage, i.e., $m_i/m_i^*$ where $m_i^*$ is fixed to investigate how the criteria change w.r.t. $m_i/m_i^*$: If the criteria decrease quickly w.r.t. $m_i/m_i^*$, it means the metric converges quickly (i.e., sample-efficient). Averaged over 5 independent trials and the error bars reflect the standard errors.

3. The MMD is computed via the analytic expression in Eq. (6) in App. A, with a radial basis function kernel $k(x, x'; \sigma) = \exp(\frac{-\|x-x'\|^2}{2\sigma^2})$ and $\sigma = 1$.

Note that the true value $\Upsilon(P_i) = -d(P_i, P^*)$ by definition.

**Correlation between obtained values and error levels.** In Fig. 6, the high $r^2$ coefficient and $p$-value (of a fitted linear regression) indicate that the valuation scores correlate with and thus can reveal the error levels. Additionally, the more extensive results on up to $n = 1000$ vendors in App. D.3.2 demonstrate a consistently high Pearson correlation coefficient between the true values (i.e., negated errors) and the measured/approximated values.

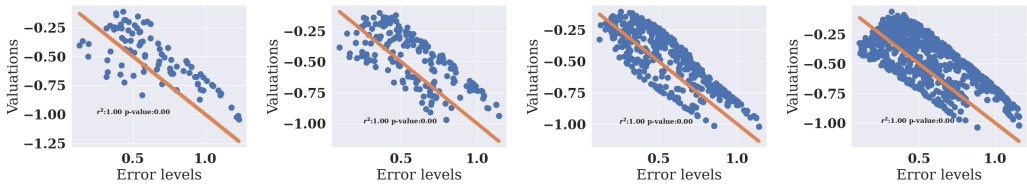

Figure 6: Valuation score vs. error level of distributions for $n = \{100, 200, 500, 1000\}$ data vendors with randomly generated data distributions. $y$-axis shows the obtained evaluation (i.e., Eq. (5)), $x$-axis shows the error level (i.e., $d(P_i, P^*)$). Orange line is a fitted linear regression, with $r^2$ coefficient and $p$-value for the significance of the independent variable (i.e., error levels): all 4 plots show $r^2$ coefficient $= 1$ and $p$-value $= 0$, indicating that the fitted linear regression (valuation regressing on error level) is statistically significant.

Table 5: Pearson correlation (mean and standard error over 10 independent trials) between true values and approximated values.

| $n\_$vendors | mean | stderr |
|---:|---:|---:|
| 5 | 0.756902 | 0.132693 |
| 10 | 0.851377 | 0.044990 |
| 20 | 0.852497 | 0.041087 |
| 100 | 0.868608 | 0.021306 |
| 200 | 0.809960 | 0.030382 |
| 500 | 0.795787 | 0.042950 |
| 1000 | 0.886822 | 0.023642 |

### D.3.3 Additional Results for Huber Setting

We present the remaining results for the Huber setting for Credit7/Credit31, MNIST/EMNIST, and MNIST/FaMNIST in Tables 6 to 8, respectively. Note that the results for $\rho(\boldsymbol{\nu}, \boldsymbol{\zeta})$ are obtained w.r.t. an available $D_{\text{val}} \sim P^*$ as the reference. Hence, our method directly uses $D_{\text{val}}$ (since it is equivalent to $D^*$ by definition) in Eq. (1) when $D_{\text{val}}$ is available. Our method uses $D_\omega$ (i.e., Eq. (3)) when $D_{\text{val}}$ is unavailable (i.e., the right columns). We highlight that Ours cond. is not applicable for when $D_{\text{val}}$ is unavailable because the conditional distribution required is not well-defined when the reference (i.e., $D_\omega$) is from a Huber distribution.

Table 6: **Classification**: $P^* = $ Credit7, $Q = $ Credit31.

| Baselines | $\rho(\boldsymbol{\nu}, \boldsymbol{\zeta})$ | $\rho(\hat{\boldsymbol{\nu}}, \boldsymbol{\zeta})$ |
|---|---|---|
| LAVA | 0.414(0.15) | 0.079(0.31) |
| DAVINZ | **0.878(0.06)** | -0.099(0.15) |
| CS | -0.813(0.03) | -0.101(0.13) |
| MMD$^2$ | 0.849(0.03) | 0.561(0.06) |
| Ours | 0.848(0.03) | **0.604(0.31)** |
| Ours cond. | 0.762(0.04) | N.A. |

Table 7: **Classification**: $P^* = $ MNIST, $Q = $ EMNIST.

| Baselines | $\rho(\boldsymbol{\nu}, \boldsymbol{\zeta})$ | $\rho(\hat{\boldsymbol{\nu}}, \boldsymbol{\zeta})$ |
|---|---|---|
| LAVA | -0.543(0.07) | 0.685(0.03) |
| DAVINZ | **0.977(0.00)** | 0.105(0.04) |
| CS | 0.760(0.08) | -0.984(0.00) |
| MMD$^2$ | 0.931(0.01) | 0.970(0.01) |
| Ours | 0.950(0.01) | **0.984 (0.01)** |
| Ours cond. | 0.971(0.01) | N.A. |

Table 8: **Classification**: $P^* = $ MNIST, $Q = $ FaMNIST.

| Baselines | $\rho(\boldsymbol{\nu}, \boldsymbol{\zeta})$ | $\rho(\hat{\boldsymbol{\nu}}, \boldsymbol{\zeta})$ |
|---|---|---|
| LAVA | -0.810(0.06) | 0.244(0.09) |
| DAVINZ | **0.864(0.05)** | 0.113(0.13) |
| CS | 0.314(0.13) | -0.810(0.03) |
| MMD$^2$ | 0.750(0.05) | **0.740(0.05)** |
| Ours | 0.747(0.05) | 0.739(0.05) |
| Ours cond. | 0.825(0.07) | N.A. |

### D.3.4 Results for Non-Huber Setting

We investigate two non-Huber settings: (1) additive Gaussian noise; (2) different supports [69], which is also generalized to an interpolated setting where the interpolation is between the different supports.

**Setting.** (1) Additive Gaussian noise: Among $n = 10$ vendors, the dataset $D_i \sim P_i := \text{MNIST} + \mathcal{N}(\mathbf{0}, \varepsilon_i \times \mathbf{I})$ where $|D_i| = m_i = 5000$ and the $\varepsilon_i$'s are $[0, 0.02, \ldots, 0.18]$. Note that though $P^* = \text{MNIST}$, each $P_i$ is *not* Huber. (2) The supports of $P^*$ that each vendor can sample from are different: on MNIST, vendor 1 only collects images of digits 0, while vendor 10 collects images of all 10 digits (called classimbalance [69]). Intuitively, vendor 10 has access to $P^*$ so its data should be the most valuable. **Results.** Tables 9 and 10 show that, when $D_{\text{val}} \sim P^*$ is available, the methods (i.e., DAVINZ, CS) that can effectively utilize $D_{\text{val}}$ can outperform our method. However, without $D_{\text{val}}$, these methods still underperform our method, especially under additive Gaussian noise. We believe it could be because these methods were not specifically designed to account for heterogeneity (which could be caused by noise), since CS performs comparably well under the class imbalance setting where the heterogeneity is due to the supports of the vendors being different instead of random noise. In particular, we find that all baselines perform sub-optimally under the additive Gaussian noise setting and when there is no clean $D_{\text{val}}$ available, which can be an interesting future direction. A possible reason is that the Gaussian noise is completely uncorrelated to the features and "destroys" the information in the data, rendering the valuation methods ineffective.

Table 9: **Non-Huber**: additive Gaussian noise.

| Baselines | $\rho(\boldsymbol{\nu}, \boldsymbol{\zeta})$ | $\rho(\hat{\boldsymbol{\nu}}, \boldsymbol{\zeta})$ |
|---|---|---|
| LAVA | -0.255(0.17) | -0.037(0.20) |
| DAVINZ | 0.848(0.02) | -0.410(0.03) |
| CS | 0.902(0.03) | -0.934(0.01) |
| MMD$^2$ | -0.085(0.04) | -0.496(0.03) |
| Ours | **0.964(0.01)** | -0.169(0.06) |
| Ours cond. | 0.892(0.04) | -0.668(0.06) |

Table 10: **Non-Huber**: classimbalance.

| Baselines | $\rho(\boldsymbol{\nu}, \boldsymbol{\zeta})$ | $\rho(\hat{\boldsymbol{\nu}}, \boldsymbol{\zeta})$ |
|---|---|---|
| LAVA | 0.439(0.26) | 0.340(0.34) |
| DAVINZ | 0.807(0.00) | 0.081(0.00) |
| CS | 0.985(0.00) | 0.871(0.01) |
| MMD$^2$ | 0.780(0.00) | -0.894(0.00) |
| Ours | 0.923(0.00) | 0.557(0.01) |
| Ours cond. | **0.989(0.00)** | **0.911**(0.00) |

**Interpolated classimbalance setting.** In addition to the "discrete" class imbalance setting, we also investigate an interpolated setting as follows: For MNIST, $n = 5, m_i = 5000$, half of $D_i$ consists of images of first $2i + 1$ digits while the other half consists of images of all 10 digits. E.g., 2500 of $D_3$ are images of digits of $0 - 6$ while the other 2500 are images of digits of $0 - 9$. Effectively, each $D_i$ contains images of *all* 10 digits, but in different proportions which increase as $i$ increases from 1 to 5. For CIFAR-10, $n = 5, m_i = 10000$ with the same interpolation implementation. Results are in Table 11. Note that for the non-Huber setting here, since the heterogeneity is only in the supports of the data (i.e., features) and not the labels, the conditional distribution for Ours cond. is indeed well-defined and thus Ours cond. is applicable here. This is different from the Huber setting examined in Sec. 5.

### D.3.5 Preliminary Experimental Results on Incentive Compatibility

We empirically demonstrate that mis-reporting decreases a vendor's value, suggesting that IC can be satisfied (approximately), as hypothesized in App. A.4.3.

Table 11: Interpolated class-imbalance setting on MNIST (left) and CIFAR-10 (right).

| Baselines | $\rho(\boldsymbol{\nu}, \boldsymbol{\zeta})$ | $\rho(\hat{\boldsymbol{\nu}}, \boldsymbol{\zeta})$ | Baselines | $\rho(\boldsymbol{\nu}, \boldsymbol{\zeta})$ | $\rho(\hat{\boldsymbol{\nu}}, \boldsymbol{\zeta})$ |
|---|---|---|---|---|---|
| LAVA | 0.459(0.24) | 0.195(0.26) | LAVA | 0.790(0.06) | 0.679(0.09) |
| DAVINZ | 0.962(0.01) | 0.706(0.04) | DAVINZ | 0.498(0.25) | 0.495(0.28) |
| CS | **0.977(0.00)** | **0.952(0.02)** | CS | **0.974(0.01)** | **0.983(0.01)** |
| MMD$^2$ | 0.839(0.05) | -0.969(0.01) | MMD$^2$ | 0.472(0.25) | 0.285(0.03) |
| Ours | 0.939 (0.02) | 0.770(0.04) | Ours | 0.931(0.11) | 0.332(0.02) |
| Ours cond. | 0.859(0.03) | 0.857(0.05) | Ours cond. | 0.846(0.04) | 0.905(0.04) |

**Settings.** Vendor $i' \in N$ is designated to mis-report: $\tilde{D}_{i'} \leftarrow D_{i'} + \mathcal{N}(0, \mathbb{I}\sigma^2)$ for $\sigma^2 = 0.2$, namely, vendor $i'$ adds zero-mean Gaussian noise to the features of the data in $D_{i'}$. This ensures $d(\tilde{P}_{i'}, P^*) > d(P_{i'}, P^*)$. For evaluation, we compute the data values (i) using Eq. (1) with a test set $D^* \sim P^*$ as the reference, denoted as the ground truth (GT); (ii) using Eq. (3) (i.e., Ours) and Tay et al. [60, Eq. (1)] (i.e., MMD$^2$), respectively, with $D_\omega$ as the reference. We include MMD$^2$ to investigate how the square affects IC (since IC of Ours can leverage the the triangle inequality of MMD that is *not* satisfied by MMD$^2$).

Corresponding to the ideal case discussed in App. A.4.3, using $D^*$ can achieve an approximate-IC with a desirable $\gamma$. In other words, GT is likely to correctly reflect the values of the vendors (including the mis-reporting $i'$). Hence, for Ours and MMD$^2$, if the data values are consistent with GT, that it suggests that (approximate) IC is more likely to be achieved.

**Results.** Fig. 7 (resp. Fig. 8) plots the average and standard error over 5 independent trials of the *change* in data values of the $n = 5$ (resp. $n = 10$) vendors as we sweep the identity of the mis-reporting vendor $i' \in \{2, 3, 4\}$.[7] Specifically, the change in data value (i.e., $y$-axis) is defined as the difference between the value of $i$ when some $i'$ (possibly $i' = i$) is mis-reporting, and that of $i$ for when no vendor is mis-reporting. IC implies a negative change (i.e., decrease) in value for the mis-reporting $i'$, which is observed for GT, Ours, and MMD$^2$ in both Figs. 7 and 8.

Note that the magnitude of the decrease for mis-reporting vendor $i'$ under Ours is more significant than that under MMD$^2$; hence it may be easier to identify the mis-reporting vendor from the truthful ones. This happens because the MMD is bounded by 1 (due to the choice of RBF kernel), and the square operation makes the value for MMD$^2$ strictly smaller in magnitude than that for MMD (i.e., Ours), as $\forall x \in (0, 1), x^2 < x$ . Therefore, MMD may be empirically more appealing than MMD$^2$.

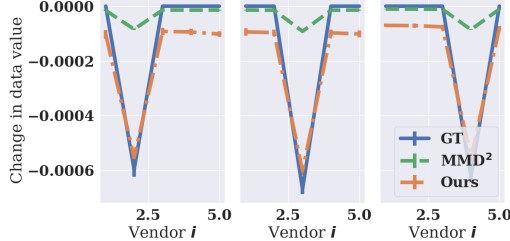
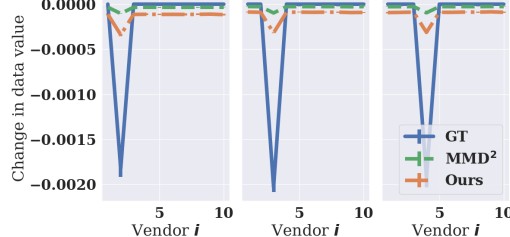

Figure 7: Change in data values for $P_{\text{MNIST}}$ and $Q_{\text{EMNIST}}$ with $n = 5$.

Figure 8: Change in data values for $P_{\text{MNIST}}$ and $Q_{\text{FaMNIST}}$ with $n = 10$.

Additional results for investigating incentive compatibility under varying settings of $P, Q$ and $n$ are presented, in Figs. 9 and 10 and Tables 12 to 15.

Figs. 9 and 10 verify the observation that the mis-reporting vendor $i'$ has a negative change (i.e., decrease) in value. Moreover, the magnitude of the decrease in value is more significant for Ours than that for MMD$^2$. The additional set of quantitative result (i.e., the Pearson correlation coefficient between the GT data values and Ours or MMD$^2$) also confirms this observation. To elaborate, take the value corresponding to $i' = 2$ and MMD$^2$ in Table 12 as an example. The Pearson coefficient (i.e.,

---

[7]The standard error bars are not visible because of the low variation across independent trials.

0.999) is between the GT data values (i.e., a vector of length $n = 5$) and the $\text{MMD}^2$ data values (i.e., a vector of length $n = 5$), under the setting that $i' = 2$ is mis-reporting. The results in Tables 12 to 15 show that the data values of both Ours and $\text{MMD}^2$ are very consistent with the GT values, importantly *without* having $D^* \sim P^*$ as the reference, thus suggesting that (approximate) IC is achievable. However, a caveat is that both methods are not very effective in achieving IC when $i' = 1$. This is precisely because $D_\omega \sim P_\omega$ is used as the reference in place of $D^* \sim P^*$ as for GT, and expected (since the discussion in App. A.4.3 suggests that the a worse approximate-IC for $P_\omega$). Specifically, IC is empirically observed when $d(P_{-i}, P^*)$ is small. When $i' = 1$, this is not satisfied, in other words $d(P_{-i}, P^*)$ would be large. This is because, $i' = 1$ is the "best" vendor in our experiment settings in that $P_{i'=1} = P^*$ (i.e., $\varepsilon_{i'=1} = 0$). Hence, if $i' = 1$ mis-reports, the remaining vendors in $N \setminus \{i'\}$ are unable to catch $i' = 1$ because the aggregate distribution $P_{-i'} = P_{-1}$ is not a very good reference (as compared to $P_{-i'}$ for $i' \neq 1$ is mis-reporting). Intuitively, if $i = 1$ is *not* mis-reporting (i.e., $i' \neq 1$), then $P_{-i'}$ contains the data from $P_1 = P^*$, and is thus a good reference.

Overall, these empirical results suggest a promising future direction of achieving (approximate) IC, by choosing a suitable metric (e.g., MMD) against a carefully constructed reference (e.g., $P_\text{U}$).

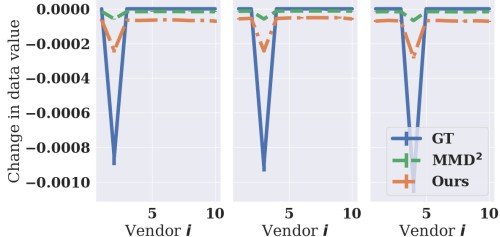 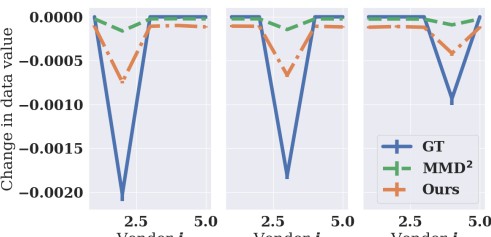

Figure 9: Change in data values for $P_\text{MNIST}$ and $Q_\text{EMNIST}$ with $n = 10$.

Figure 10: Change in data values for $P_\text{MNIST}$ and $Q_\text{FaMNIST}$ with $n = 5$.

Table 12: Average and standard error (over $5$ independent trials) of Pearson coefficients with GT for $P_\text{MNIST}$ and $Q_\text{EMNIST}$ with $n = 5$, rows $i' = 2, 3, 4$ corresponding to Fig. 7.

| $i'$ | $\text{MMD}^2$ | Ours |
|---|---|---|
| 1 | 0.485 (0.03) | 0.476 (0.03) |
| 2 | 0.999 (0.00) | 0.999 (0.00) |
| 3 | 1.000 (0.00) | 1.000 (0.00) |
| 4 | 1.000 (0.00) | 1.000 (0.00) |
| 5 | 1.000 (0.00) | 1.000 (0.00) |

Table 13: Average and standard error (over $5$ independent trials) of Pearson coefficients with GT for $P_\text{MNIST}$ and $Q_\text{FaMNIST}$ with $n = 5$, rows $i' = 2, 3, 4$ corresponding to Fig. 10.

| $i'$ | $\text{MMD}^2$ | Ours |
|---|---|---|
| 0 | 0.515 (0.01) | 0.506 (0.01) |
| 1 | 1.000 (0.00) | 1.000 (0.00) |
| 2 | 0.999 (0.00) | 0.999 (0.00) |
| 3 | 0.999 (0.00) | 0.999 (0.00) |
| 4 | 0.997 (0.00) | 0.998 (0.00) |

### D.4 Observed Linear Scaling w.r.t. $n$ and $m$

We demonstrate the scalability of our method w.r.t. the number $n$ of vendors and the sample size $m$, in terms of execution time and memory (RAM and GPU).

**Plots showing linear scaling.** Fig. 11 observes linear scaling between time vs. $m_i$ (top left), time vs. $n$ (bottom left), RAM vs. $m_i$ (top right) and RAM vs. $n$ (bottom right). Crucially, this helps ensure the practical applicability of our method in terms of implementation and execution.

Table 14: Average and standard error (over 5 independent trials) of Pearson coefficients with GT for $P_{\text{MNIST}}$ and $Q_{\text{EMNIST}}$ with $n = 10$, rows $i' = 2, 3, 4$ corresponding to Fig. 9.

| $i'$ | MMD$^2$ | Ours |
|---|---|---|
| 0 | 0.386 (0.01) | 0.383 (0.01) |
| 1 | 0.997 (0.00) | 0.997 (0.00) |
| 2 | 0.997 (0.00) | 0.997 (0.00) |
| 3 | 0.999 (0.00) | 0.999 (0.00) |
| 4 | 0.998 (0.00) | 0.998 (0.00) |
| 5 | 0.998 (0.00) | 0.998 (0.00) |
| 6 | 0.999 (0.00) | 0.999 (0.00) |
| 7 | 0.999 (0.00) | 0.999 (0.00) |
| 8 | 0.998 (0.00) | 0.998 (0.00) |
| 9 | 0.993 (0.00) | 0.994 (0.00) |

Table 15: Average and standard error (over 5 independent trials) of Pearson coefficients with GT for $P_{\text{MNIST}}$ and $Q_{\text{FaMNIST}}$ with $n = 10$, rows $i' = 2, 3, 4$ corresponding to Fig. 8.

| $i'$ | MMD$^2$ | Ours |
|---|---|---|
| 1 | 0.364 (0.01) | 0.362 (0.01) |
| 2 | 0.999 (0.00) | 0.999 (0.00) |
| 3 | 0.997 (0.00) | 0.997 (0.00) |
| 4 | 0.999 (0.00) | 0.998 (0.00) |
| 5 | 0.998 (0.00) | 0.998 (0.00) |
| 6 | 0.995 (0.00) | 0.995 (0.00) |
| 7 | 0.996 (0.00) | 0.995 (0.00) |
| 8 | 0.954 (0.01) | 0.950 (0.01) |
| 9 | 0.852 (0.06) | 0.858 (0.05) |
| 10 | 0.997 (0.00) | 0.998 (0.00) |

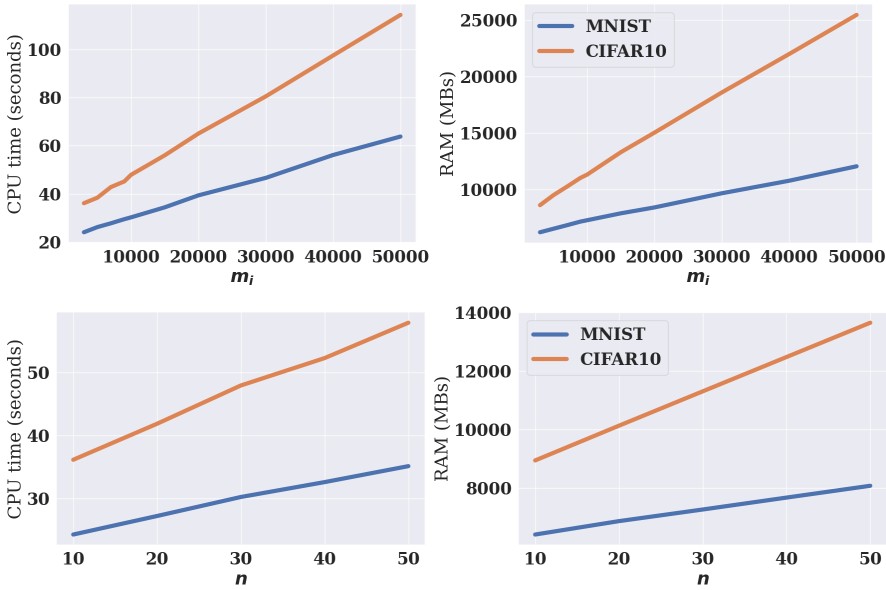

Figure 11: Top: time and peak memory vs. $m_i$ at $n = 30$; Bottom: time and peak memory vs. $n$ at $m = 10000$. MNIST or CIFAR10 denotes the dataset used.

**RAM, CUDA and CPU time results.** We include the detailed results for RAM, CUDA, and time on MNIST (Tables 19 to 21) and CIFAR10 (Tables 16 to 18), respectively.

Table 16: CUDA Memory in MBs for CIFAR10.

| $m_i \backslash n$ | 10 | 20 | 30 | 40 | 50 |
|---|---|---|---|---|---|
| 3000 | 140.598272 | 182.904320 | 222.904320 | 264.693248 | 304.847360 |
| 5000 | 141.556736 | 182.904320 | 222.904320 | 264.693248 | 304.847360 |
| 7000 | 143.563776 | 182.904320 | 222.904320 | 264.693248 | 304.847360 |
| 9000 | 144.983040 | 182.904320 | 222.904320 | 264.693248 | 304.847360 |
| 10000 | 145.862144 | 182.904320 | 222.904320 | 264.693248 | 304.847360 |
| 15000 | 150.258176 | 182.904320 | 222.904320 | 264.693248 | 304.847360 |
| 20000 | 155.048960 | 182.904320 | 222.904320 | 264.693248 | 304.847360 |
| 30000 | 164.999168 | 189.000192 | 222.904320 | 264.693248 | 304.847360 |
| 40000 | 173.798400 | 205.969920 | 238.375936 | 271.930368 | 304.847360 |
| 50000 | 182.904320 | 222.904320 | 264.693248 | 304.847360 | 346.482176 |

Table 17: RAM in MBs for CIFAR10.

| $m_i \backslash N$ | 10 | 20 | 30 | 40 | 50 |
|---|---|---|---|---|---|
| 3000 | 7726.480469 | 8022.242188 | 8602.214844 | 8813.792969 | 9256.371094 |
| 5000 | 8036.699219 | 8683.101562 | 9489.097656 | 10134.582031 | 10780.066406 |
| 7000 | 8318.515625 | 9469.835938 | 10239.902344 | 11060.789062 | 11963.085938 |
| 9000 | 8628.960938 | 9815.738281 | 11016.425781 | 12092.296875 | 13129.046875 |
| 10000 | 8945.777344 | 10132.257812 | 11306.156250 | 12478.398438 | 13650.953125 |
| 15000 | 9444.472656 | 11187.527344 | 13286.566406 | 14993.132812 | 16758.398438 |
| 20000 | 10058.050781 | 12478.628906 | 15021.453125 | 17166.140625 | 19696.039062 |
| 30000 | 11374.105469 | 15036.773438 | 18574.199219 | 22034.144531 | 25492.121094 |
| 40000 | 12545.460938 | 17267.832031 | 21975.371094 | 26613.242188 | 31401.500000 |
| 50000 | 13877.929688 | 19598.089844 | 25435.593750 | 31254.929688 | 37299.406250 |

Table 18: CPU time in seconds for CIFAR10.

| $m_i \backslash N$ | 10 | 20 | 30 | 40 | 50 |
|---|---|---|---|---|---|
| 3000 | 40.075573 | 33.853308 | 36.099317 | 36.748012 | 38.443388 |
| 5000 | 32.896979 | 35.561858 | 38.386079 | 41.070452 | 44.552341 |
| 7000 | 34.791889 | 38.024787 | 42.796535 | 45.675799 | 50.420825 |
| 9000 | 37.090735 | 40.622674 | 45.164939 | 50.043854 | 55.225169 |
| 10000 | 36.123430 | 41.834278 | 47.907592 | 52.253957 | 57.867871 |
| 15000 | 38.382908 | 47.813846 | 56.015236 | 63.631425 | 72.990464 |
| 20000 | 42.548813 | 54.514572 | 65.067422 | 76.599092 | 89.003306 |
| 30000 | 46.957031 | 65.156033 | 80.534339 | 97.402354 | 116.374963 |
| 40000 | 52.663162 | 76.029116 | 97.542482 | 120.323142 | 144.956402 |
| 50000 | 59.434006 | 89.602017 | 114.438738 | 146.690117 | 173.507339 |

**Scalability comparison against DAVINZ.** The implementation of DAVINZ also includes an MMD computation (similar to our proposed method), but additionally linearly combined with a the neural tangent kernel (NTK)-based score. While in some cases (e.g., Table 3), DAVINZ and our proposed method perform comparably, we highlight that our method is more scalable. The main reason is that the gradient computation from NTK in DAVINZ requires additional memory, specifically CUDA memory due to leveraging GPU for gradient computation. See Table 22 for a scalability experiment for DAVINZ with $n = 10$ on MNIST with a standard convolutional neural network used for all MNIST-related experiments in this work.

In contrast, under the same setting of $n = 10$ data vendors each with $m_i = 10000$ samples, our method requires less than $0.1$ GBs of CUDA memory (see the fifth row, first column of Table 19. Note that we were not able to collect results for DAVINZ on more extensive settings due to hardware limitations (i.e., larger values for $n$ and $m_i$ leads to out-of-memory errors on our standard GPUs with

Table 19: CUDA memory for MNIST.

| $m_i \backslash N$ | 10 | 20 | 30 | 40 | 50 |
|---|---|---|---|---|---|
| 3000 | 95.339008 | 138.832384 | 178.832384 | 220.621312 | 260.775424 |
| 5000 | 97.097728 | 138.832384 | 178.832384 | 220.621312 | 260.775424 |
| 7000 | 98.855936 | 138.832384 | 178.832384 | 220.621312 | 260.775424 |
| 9000 | 101.214720 | 138.832384 | 178.832384 | 220.621312 | 260.775424 |
| 10000 | 101.693952 | 138.832384 | 178.832384 | 220.621312 | 260.775424 |
| 15000 | 105.895936 | 138.832384 | 178.832384 | 220.621312 | 260.775424 |
| 20000 | 110.686720 | 138.832384 | 178.832384 | 220.621312 | 260.775424 |
| 30000 | 119.679488 | 143.097856 | 178.832384 | 220.621312 | 260.775424 |
| 40000 | 130.443776 | 162.443776 | 195.455488 | 227.238400 | 260.775424 |
| 50000 | 138.832384 | 178.832384 | 220.621312 | 260.775424 | 302.410240 |

Table 20: RAM in MBs for MNIST.

| $m_i \backslash N$ | 10 | 20 | 30 | 40 | 50 |
|---|---|---|---|---|---|
| 3000 | 6077.753906 | 6148.613281 | 6204.640625 | 6332.785156 | 6478.375000 |
| 5000 | 6093.554688 | 6284.605469 | 6517.027344 | 6718.703125 | 6934.570312 |
| 7000 | 6122.351562 | 6535.925781 | 6828.468750 | 7151.273438 | 7438.753906 |
| 9000 | 6206.800781 | 6694.765625 | 7148.257812 | 7543.253906 | 7876.445312 |
| 10000 | 6414.144531 | 6872.023438 | 7269.164062 | 7676.304688 | 8080.292969 |
| 15000 | 6665.367188 | 7210.253906 | 7880.492188 | 8484.843750 | 9024.613281 |
| 20000 | 6867.117188 | 7651.234375 | 8402.281250 | 9231.320312 | 9992.460938 |
| 30000 | 7205.328125 | 8409.937500 | 9653.914062 | 10725.601562 | 11970.367188 |
| 40000 | 7559.824219 | 9132.652344 | 10764.582031 | 12254.757812 | 13927.617188 |
| 50000 | 8096.402344 | 9891.027344 | 12040.925781 | 13953.308594 | 15842.777344 |

Table 21: CPU time in seconds for MNIST.

| $m_i \backslash N$ | 10 | 20 | 30 | 40 | 50 |
|---|---|---|---|---|---|
| 3000 | 45.675888 | 23.173012 | 24.010044 | 24.685510 | 25.532540 |
| 5000 | 22.843993 | 24.338142 | 26.162306 | 27.275437 | 28.386587 |
| 7000 | 23.884181 | 25.619524 | 27.735549 | 29.385311 | 31.042997 |
| 9000 | 23.949323 | 26.902086 | 29.450777 | 31.600939 | 33.862759 |
| 10000 | 24.259586 | 27.199071 | 30.215128 | 32.577134 | 35.111519 |
| 15000 | 28.153503 | 30.075059 | 34.379974 | 38.391948 | 42.446352 |
| 20000 | 27.595278 | 34.082700 | 39.362670 | 46.231271 | 49.121318 |
| 30000 | 30.255474 | 38.232212 | 46.626252 | 54.726982 | 63.040075 |
| 40000 | 33.875198 | 44.688145 | 56.146309 | 66.236627 | 77.427208 |
| 50000 | 35.978749 | 50.792472 | 63.822176 | 77.274825 | 92.584505 |

10 GBs of memory and the official implementation [67, Github repo] does not implement a way to take advantage of multiple GPUs (if available) to distribute the CUDA memory load.

Table 22: Maximum CUDA, RAM and time for DAVINZ with $n = 10$ data vendors on MNIST with a standard convolutional neural network.

| $m_i$ | maximum CUDA (in MBs) | RAM (in MBs) | CPU time (in seconds) |
|---|---|---|---|
| 3000 | 4885.189 | 5201.132 | 50.419 |
| 5000 | 4885.189 | 6840.75 | 44.173 |
| 7000 | 4885.189 | 6843.371 | 46.229 |
| 9000 | 4885.189 | 6836.921 | 51.436 |
| 10000 | 5523.673 | 6843.773 | 50.491 |
| 15000 | 5523.673 | 6926.535 | 56.739 |
| 20000 | 7118.552 | 7358.734 | 68.144 |
| 30000 | 7118.552 | 8331.660 | 86.353 |
| 40000 | 7323.634 | 9154.773 | 107.132 |
| 50000 | 7323.634 | 10053.316 | 132.206 |

