# OpenReview forum: "Data Distribution Valuation"
_NeurIPS.cc/2024/Conference — NeurIPS 2024 poster_

### Official Review · Reviewer_Sj3P · 2024-06-23

**Soundness:** 3
**Presentation:** 3
**Contribution:** 3
**Rating:** 6
**Confidence:** 4

**Summary:**

This paper starts by highlighting the importance of accurately assessing the value of data distributions, especially in the growing data economy. To address the problem, this paper introduces a data distribution valuation method based on Maximum Mean Discrepancy (MMD) for comparing the value of data distributions from samples. This paper assumes that each vendor $i$ holds a distribution $P_i$ which follows a Huber model, and is a mixture of ground truth distribution $P^*$ and arbitrary distribution $Q_i$. Based on this assumption, the authors discuss the theoretical foundations and assumptions, providing detailed proofs and derivations for the proposed methods. The study addresses heterogeneity in data distributions and the challenges of combining multiple data vendors' datasets. Experimental results demonstrate the sample efficiency and effectiveness of the proposed methods in ranking data distributions.

**Strengths:**

1.	This paper provides a detailed explanation of the theoretical foundations for the valuation of data distributions, including assumptions and proofs, ensuring the rigor and completeness of the theory.

2.	This paper studies a meaningful problem: how to compare the values of data distributions from their samples, which can help evaluate the value of data provided by different vendors.

3.	This paper is well-organized and easy to follow. The paper introduces a novel method based on maximum mean discrepancy (MMD) for data distribution valuation, offering a fresh perspective in the field.

**Weaknesses:**

1.	This paper relies on certain assumptions, such as the Huber model of data heterogeneity, which may not always hold in real-world scenarios.

2.	The experiment for ranking data distributions lacks generalizability.

3.	Using data samples to represent data distribution can cause issues, such as dealing with malicious data vendors.

**Questions:**

1.	In real-world scenarios, data distributions often encompass various complex heterogeneity factors that the Huber model may not accurately capture. Data collected from different sources often exhibit significant variability and may not follow the same distribution.

2.	The experiment in the article for ranking data distributions involves mixing two similar datasets, which fails to adequately represent the heterogeneity between datasets provided by different vendors, thus affecting the generalizability of the results.

3.	In real-world scenarios, some vendors may provide data samples that do not accurately reflect their true data distribution. Some malicious vendors might even forge samples (for example, using some real data as samples while the rest of dataset is synthetic or useless data). The article does not take this into account.

---

> ### Author Rebuttal · Authors · 2024-08-06
>
> We thank Reviewer Sj3P for reviewing our paper, and appreciating the theoretical rigor of our work, the meaningfulness of our studied problem and the novelty and fresh perspective of our method.
>
> We would like to address the comments and feedback as follows.
>
> > This paper relies on certain assumptions, such as the Huber model of data heterogeneity, which may not always hold in real-world scenarios. In real-world scenarios, data distributions often encompass various complex heterogeneity factors that the Huber model may not accurately capture. Data collected from different sources often exhibit significant variability and may not follow the same distribution.
>
> We acknowledge that the assumption on the Huber model presents a theoretical limitation (line 404), and we have investigated settings where the Huber model is _not_ satisfied (i.e., additive Gaussian noise, and a class imbalance setting) where our method continues to perform well, and this is mentioned in Section 3 (lines 138-139) and Section 6 (lines 405-406). These results are deferred to the Appendix D.3.3 due to page constraints. In particular, the class imbalance setting represents the observation that "Data collected from different sources often exhibit significant variability and may not follow the same distribution."
>
> Nevertheless, we wish to point out that this assumption of Huber model is to ensure the theoretical tractability of analysis (e.g., Proposition 1, Equation 2), which has not been explored previously in the sense that prior works have not considered a precise form of data distribution or characterization of data heterogeneity (elaborated in Appendix B.3 in lines 909-930). Additionally, we discuss the considerations of extending beyond the Huber model (lines 1110-1114) and the opportunities for future research.
>
> > The experiment for ranking data distributions lacks generalizability. The experiment in the article for ranking data distributions involves mixing two similar datasets, which fails to adequately represent the heterogeneity between datasets provided by different vendors, thus affecting the generalizability of the results.
>
> In our main paper, the settings for the empirical results follow the Huber model, to verify our theoretical results and method that are based on the Huber model. We acknowledge that a single formalization of data heterogeneity cannot generalize to all settings (lines 58-60), so we have also performed experiments under _non-Huber_ settings in Appendix D.3.3. To elaborate, we investigate two specific non-Huber settings for data heterogeneity: (1) with additive Gaussian noise and (2) class imbalance setting. For (1), the datasets are "perturbed" with additive Gaussian noise in the features. For (2), different data vendors are restricted to observe different supports of the full data distribution, which is a commonly adopted setting for data heterogeneity in distributed machine learning [68]. We also consider a continuous interpolation of (2). In these settings, our method continues to perform well, as shown in Tables 8,9 & 10 in Appendix D.3.3. Nevertheless, we acknowledge that these settings do not cover all cases of data heterogeneity in practice; extending the method and results to more settings is an important direction for future work.
>
> > Using data samples to represent data distribution can cause issues, such as dealing with malicious data vendors. In real-world scenarios, some vendors may provide data samples that do not accurately reflect their true data distribution. Some malicious vendors might even forge samples (for example, using some real data as samples while the rest of dataset is synthetic or useless data). The article does not take this into account.
>
> Indeed the consideration of incentivizing the data vendors to truthfully report their samples is of great practical importance, and it has difficulties especially in our setting which does _not_ assume to have access to a ground truth reference distribution. Formally, the notion of _incentive compatibility_ of a valuation metric can ensure that the vendors perform truthful reporting. In other words, it deters malicious vendors or discourages vendors from reporting forged samples. In our setting of data distribution valuation, if the valuation is accurate (e.g., with access to the ground truth reference distribution such as in Section 4.1), it can be shown to satisfy incentive compatibility; and if the valuation has a bounded error (e.g., without the ground truth reference distribution, but with some approximated reference distribution such as in Section 4.2), then the valuation can be said to satisfy an approximate (i.e., weaker) version of incentive compatibility. We make these precise in Appendix A.4.3 (lines 764-795), and provide preliminary empirical results in Appendix D.3.4. In our empirical investigation, the data vendor who mis-reports the samples receives a significantly lower value (Figures 6,7,8 & 9 in Appendix D.3.4). These results demonstrate that our proposed method can mitigate the issue of malicious data vendors to some extent. This discussion and the results are deferred to appendix due to page constraints and that these are not the primary focus of our work. We point out that our discussion and results are preliminary in this very important direction, which may require a separate and in-depth treatment, and hope that our preliminary results can serve as starting points for future research.
>
> We wish to thank Reviewer Sj3P for the positive feedback and questions. We hope our response has clarified your questions and helped improve your opinion of our work.

---

### Official Review · Reviewer_s86g · 2024-07-13

**Soundness:** 2
**Presentation:** 2
**Contribution:** 3
**Rating:** 6
**Confidence:** 5

**Summary:**

The valuation of data is crucial in data marketplaces. Instead of assessing the value of a specific dataset, this paper focuses on the valuation of data distribution behind the dataset itself. For example, several vendors are trying to sell different or even the same datasets, what is the best distribution to purchase when we only observe a sampled dataset? The authors model the problem of data distribution valuation and use the MMD-based method as a metric to evaluate the valuation. They provide theoretically guaranteed policies for buyers to take action and empirically demonstrate it using real-world datasets. The results indicate that the method is sample-efficient and outperforms other valuation metrics.

**Strengths:**

* Originality: This paper is interesting as it evaluates the distribution behind a sampled dataset rather than the dataset itself. This approach is new and can be valuable when dealing with partially sampled datasets.
* Quality: The results seem promising and support the use of MMD-based metrics for assessing data distribution.
* Clarity: The writing and formulation are clear and easy to understand.
* Significance: In data marketplaces, most data is only available for preview and is often sampled. Understanding the value of data distribution is beneficial for the field and users.

**Weaknesses:**

1. The paper's motivation is interesting and contributes to the field. However, I believe there is a missing experiment regarding ranking different error levels of distributions. Suppose we have five distributions ranging from 100% correct to 0% correct. Can the valuation score accurately rank these distributions or reveal their actual error levels? This experiment differs from directly comparing the valuation of the dataset itself. We should see a regression line, where the x-axis is the error level of distribution, while the y-axis is the valuation score.
2. Why do the correlation scores drop when we move the dataset from CIFAR10 to CIFAR100? The description about this is not clear to me. More clarification on this will be helpful.
3. The method of sampling from a distribution to create a dataset can influence its evaluation. Have there been any empirical findings or methods to address sampling bias? This missing experiment can justify the robustness of the valuation function.

**Questions:**

1. The method used to sample from a distribution to construct a dataset can affect its valuation. Are there any empirical results or methods to overcome sampling bias?
2. How does the accuracy of valuation change when a large number of vendors contribute to the mixed reference distribution? Also, does the valuation score reveal the relative level of two distributions or their absolute values?
3. In Equation 1, what is the reason for giving the value function a negative term instead of taking the reciprocal?

**Limitations:**

I didn't see any potential negative societal impact of their work.

---

> ### Author Rebuttal · Authors · 2024-08-07
>
> We thank Reviewer s86g for reviewing our paper, and for the positive feedback on the novelty of our approach, the quality of our methodology and solution, the clarity of our writing and significance of our work.
>
> We wish to provide the following clarifications. The requested experimental results (and detailed settings) are in `response.pdf` in the global comment.
>
> > ... We should see a regression line, where the x-axis is the error level of distribution, while the y-axis is the valuation score.
>
> Figure 11 in `response.pdf` shows a relatively clear negative correlation.
>
> > Can the valuation score accurately rank these distributions or reveal their actual error levels?"
>
> The high $r^2$ coefficient and the $p$-value (of a fitted linear regression) indicate that the valuation scores correlate with and thus can reveal the error levels. Table 22 in `response.pdf` shows more extensive and quantitative results.
>
> Regarding settings, we follow the reviewer's suggestions as closely as possible:
> - We consider up to $1000$ vendors for generalizability.
> - The error level is by $d(P_i,P^*)$ where a low (resp. high) MMD means a low (resp. high) error, instead of the percentage-type error levels since it is not precisely defined for a distribution to be 60\% correct. Also, our main paper (e.g., Table 2) already shows a high correlation between valuation score and machine learning accuracy.
> - We obtain the value from distributions directly, using discrete distributions and an analytic expression of MMD (from lines 547-575). This is in general not possible due to not knowing the exact analytic pdfs of data distributions (e.g., MNIST).
>
> > Why do the correlation scores drop when we move the dataset from CIFAR10 to CIFAR100? The description about this is not clear to me. More clarification on this will be helpful.
>
> We clarify that the first two columns of Table 2 are _not_ showing results of "moving from CIFAR10 to CIFAR100". In our setting for Table 2, CIFAR10 is designated as $P^*$ and CIFAR100 is designated as $Q$ (lines 311-315, or Table 4 in Appendix D.2). The first (resp. second) column of Table 2 shows the correlation when a clean hold-out validation set $D_{\text{val}}\sim P^*$ is available (resp. unavailable). The results (Tables 2 & 3) show that the effectiveness of a valuation metric decreases without $D_{\text{val}}$. Importantly, Ours (and MMD$^2$) experience a smaller decrease/"drop" and are thus preferrable in practice where $D_{\text{val}}$ is unavailable.
>
> > The method used to sample from a distribution ... can affect its valuation. Are there any empirical results ... to overcome sampling bias?
>
> For a fixed distribution, different sampling methods yield different sampled datasets, and thus different valuations. This is an expected and correct behavior, and in particular, the "negative" sampling bias can be mitigated by our proposed valuation.
>
> A "positive" sampling bias is when a data vendor, using knowlegde of $P^*$, to prioritize the sampling from $P^*$ in their $D_i$. This will result in a higher valuation. It is a correct behavior as $D_i$ contains more information about $P^*$ (i.e., the ground truth). A "negative" sampling bias is when the vendor performs sampling in a way that "moves away" from $P^*$ (e.g., injecting noise), then this will result in a lower valuation, as $D_i$ contains less information about $P^*$.
>
> In practice, the positive bias is rare due to not knowing $P^*$. However, the negative bias is possible, can deteriorate the quality of the data and should be discouraged. This is precisely formalized via the so-called incentive compatibility (from mechanism design), which we investigate w.r.t. our valuation, both theoretically (Appendix A.4.3, lines 764-795) and empirically (Appendix D.3.4, lines 1235-1281). In particular, the empirical results (e.g., Table 6) verify that a vendor who injects such a negative bias by misreporting (noisy data) receives a lower valuation, demonstrating that the negative sampling bias can be mitigated.
>
> > How does the accuracy of valuation change when a large number of vendors ... ?
>
> Table 22 shows that the accuracy of valuation (measured as the Pearson correlation between obtained values and the unknown true values) remains high for up to $1000$ data vendors.
>
> > Also, does the valuation score reveal the relative level of two distributions or their absolute values?
>
> The valuation score reveals a relative level. It can be combined with additional transformations (e.g., translation): Let $d(P_i,P^*)=0.1$, $d(P_j,P^*)=0.2$, so their values are $\Upsilon(P_i) = -0.1$ and $\Upsilon(P_j)=-0.2$. To obtain non-negative values, we can linearly translate by $+1$ gives $\Upsilon(P_i)+1 = -0.1+1 = 0.9$ and $\Upsilon(P_j)+1=-0.2+1=0.8$. Note the absolute value of MMD is bounded by the upper bound $K$ of the kernel $k$ (e.g., $K=1$ for the RBF kernel), so a translation by $+K$ ensures that all values are non-negative. While additional transformations are possible (e.g., positive scaling after translation), the transformed values should be interpreted in the specific use case that motivates the transformation.
>
> > In Equation 1, what is the reason for giving the value function a negative term instead of taking the reciprocal?
>
> Taking the negation has theoretical and practical reasons. Theoretically, it (i) preserves the triangle inequality of MMD (e.g., used in Prop. 1), and (ii) enables the precise characterization of heterogeneity in Equation 2 (lines 191-194). Practically, (i) the uniform convergence of the MMD estimator (Lemma 1 in Appendix A.2) continues to apply (which may not for the reciprocal), making the value easy to estimate, while (ii) taking the reciprocal may lead to complications (e.g., division-by-zero errors).
>
> We thank Reviewer s86g for the comments and questions, and will incorporate the discussion and additional results in our revision. We hope that our response has clarified your questions and helped raise your opinion of our work.

---

> > ### Comment · Reviewer_s86g · 2024-08-09
> >
> > Thanks for conducting additional experiments and clearing my concerns on sampling, more vendors, and different error levels' valuations. Regression lines in your attachment have a negative correlation and enhance the quality of this work. Thanks for clarifying my questions on cifar10 and cifar100. I have increased my rating and support this paper. Thanks for your response!

---

> > > ### Author Response · Authors · 2024-08-09
> > > **Thank you for the acknowledgement and increasing your score**
> > >
> > > We wish to thank Reviewer s86g for acknowledging our rebuttal and increasing the score. We really appreciate your support!

---

### Official Review · Reviewer_YGYg · 2024-07-13

**Soundness:** 3
**Presentation:** 3
**Contribution:** 3
**Rating:** 7
**Confidence:** 4

**Summary:**

This paper addresses the problem of data distribution valuation in data markets, where buyers need to evaluate the quality of data distributions to make informed purchasing decisions. The authors formulate the problem and identify three technical challenges: heterogeneity modeling, defining the value of a sampling distribution, and choosing a reference data distribution. They make three key design choices: assuming a Huber model for data heterogeneity, using negative maximum mean discrepancy (MMD) as the value metric, and considering a class of convex mixtures of vendor distributions as the reference. The paper derives an error guarantee and comparison policy for the proposed method and demonstrates its effectiveness on real-world classification and regression tasks. Overall, this work provides a novel framework for data distribution valuation, enabling buyers to make informed decisions in data markets.

**Strengths:**

1. The paper introduces a novel approach to data valuation by focusing on the value of the underlying data distribution from a small sample. This addresses a gap in existing methods, which typically do not formalize the value of sampling distributions or provide actionable policies for comparing them.


2. The paper employs a Huber model to capture data heterogeneity and utilizes the maximum mean discrepancy (MMD) for evaluating sampling distributions. This combination allows for precise, theoretically grounded comparisons of data distributions and allows for sample efficient assessment of the valuation. Assuming a convex combination of the distribution as the reference they provide an error guarantee without making the common assumption of knowing the reference distribution.


3. The authors validate their method through real-world classification and regression tasks. The demonstrated sample efficiency and effectiveness of their MMD-based valuation method, particularly its superior performance in most classification settings compared to existing metrics highlights the practical relevance and robustness of their approach.

**Weaknesses:**

1. Based on Theorem 1 the valuation of $D$ boils down to the samples available from it and the averaged out heterogeneity $d(Q_\omega, P^*)$. In practice, a small sample that looks cleaner could be preferred over a bigger but noisy sample. It looks like the current results do not account for individual noise (heterogeneity), I see it in the Huber model but the theoretical results are averaging out as in Observation 1.

2. The sample complexity of $O(\frac{1}{\sqrt{m}})$ to estimate MMD, might even be sufficient for the learning task at hand, so why would vendors be willing to show such a big sample of the dataset and by seeing samples from each vendor won’t one be able to accomplish the learning objective without even selecting a vendor and buying a larger sample from them.

**Questions:**

See above.

**Limitations:**

Yes.

---

> ### Author Rebuttal · Authors · 2024-08-06
>
> We thank Reviewer YGYg for reviewing our paper, and for appreciating the novelty of our approach and acknowledging our theoretical and empirical results.
>
> We would like to respond to the feedback and comments as follows,
>
> > Based on Theorem 1 the valuation of $D$ boils down to the samples available from it and the averaged out heterogeneity $d(Q_\omega, P^*)$. In practice, a small sample that looks cleaner could be preferred over a bigger but noisy sample. It looks like the current results do not account for individual noise (heterogeneity), I see it in the Huber model but the theoretical results are averaging out as in Observation 1.
>
> We point out that our definition (Equation 1) enables a precise characterization of the individual noise (w.r.t. $\epsilon_i, Q_i$) as in Equation 2 and in lines 190-196. Nevertheless, we acknowledge that our method leveraging the "averaging" and Observation 1 is not guaranteed to precisely identify the individual noise (e.g., by directly obtaining $\epsilon_i, Q_i$). We wish to highlight that this may be theoretically difficult to do (i.e., precisely and quantitatively identifying the individual noise), because we do _not_ make the assumption of knowing the reference distribution (as recognized by the reviewer). Intuitively, without knowing what the ground truth is (i.e., the reference distribution), one cannot reliably identify the individual noise, because it is difficult to distinguish noise from signal. We provide a more technical elaboration on this perspective and highlight the theoretical difficulties in Appendix C (see Q6 in lines 1054-1087) with corroborating observations and results from other fields (i.e., robust statistics and mechanism design). Informally, our discussion implies that additional assumptions are needed in order to identify, account for or manage individual noise precisely and quantitatively, so exploring the exact form of such assumptions and the methods to do so is an important future direction (in Appendix B.3 in lines 951-977 and also in Appendix C in lines 1084-1087).
>
> > The sample complexity of $\mathcal{O}(1/\sqrt{m})$ to estimate MMD, might even be sufficient for the learning task at hand, so why would vendors be willing to show such a big sample of the dataset and by seeing samples from each vendor won’t one be able to accomplish the learning objective without even selecting a vendor and buying a larger sample from them.
>
> The vendors may _not_ need to show a big sample of the dataset. As an example, the actual data need not be disclosed (in plaintext) by utilizing cryptographic approaches, such as computing MMD in a trusted execution environment (TEE). This provides a verifiable computed MMD _without_ disclosing the data. To expand on this, the vendors may each provide a relatively small public preview sample (e.g., a few hundred data points) and the interested buyers can request a "quotation" computed from a much larger sample dataset from the vendor in a cryptographically secure way: (i) buyer does not see the dataset; (ii) buyer can verify that the computation is correct. We note that our work aims to provide the theoretical foundations for an MMD-based data valuation, so the design for such cryptographic computations is beyond the scope of our current work.
>
> To a prospective buyer, the computed MMD (e.g., in a TEE) with the sample size, can provide useful information to gauge the error margin for estimating the value of the data, using the sample complexity results as a guideline. Concretely, suppose that vendor $i$'s value is estimated to be $0.8$ with an error margin of $0.2$ while vendor $j$'s value is estimated to be $0.7$ with an error margin of $0.05$ because vendor $j$ supplied a larger sample. A risk-averse buyer (who cares about the worst-case scenario) may pick vendor $j$ over $i$ since the lower bound of the value for $j$ is $0.7-0.05=0.65$, greater than that ($0.8-0.2=0.6$) for vendor $i$.
>
> That being said, in many operational data marketplaces and suppliers (e.g., Datarade, Snowflake, or the Bloomberg terminal), their business models may already account for this possible scenario and factor it into their pricing mechanism, since after all, not every buyer who expresses interest in the preview sample would eventually purchase the product (i.e., data distribution). We think that the buyers who end up purchasing the data would have stricter error requirements than what the preview sample can meet. We will incorporate the above discussion in our revision.
>
> We thank Reviewer YGYg for the detailed feedback and hope that we have addressed your comments and improved your opinion of our work.

---

> > ### Comment · Reviewer_YGYg · 2024-08-08
> >
> > Thank you for the rebuttal and congratulations for the nice work! I have increased my score.
> > I presumed that MMD will be computed on the sample shown to the buyer. It would be nice to have some discussion in the paper to avoid this confusion.
> >
> > Minor stuff: please improve the presentation of results in the tables.
> >
> > You don't have to respond to this. I am curious about, how vendors can create the samples without revealing too much. In particular in your setting, as a buyer I can collect the freely visible samples from multiple vendors and train a model. If the vendor distributions are diverse enough then a buyer would still be able to get a very good model from those free samples (freeloader, lol). I am curious about when such freeloading is possible and when it is not, theoretically and can the vendors do something to prevent it. If it requires them to co-ordinate that might introduce conflicts.

---

> > > ### Author Response · Authors · 2024-08-09
> > > **Thank you for increasing the score**
> > >
> > > We would like to thank Reviewer YGYg for acknowledging our rebuttal and increasing the score. We will definitely take note of your feedback and incorporate it into our revision.
> > >
> > > As for your question on if/"how vendors can create the samples without revealing too much", and if it requires coordination among vendors and its implications, we believe it to be an interesting avenue to explore an algorithmic solution intersecting statistics/machine learning and game theory and will definitely note this in our revision.
> > >
> > > Thanks again for your encouraging feedback and helpful comments.

---

### Author Rebuttal · Authors · 2024-08-07

We wish to thank the reviewers for reviewing our paper and providing the detailed feedback. We especially appreciate the positive feedback on the __novelty of our work__ (all reviewers), the __clarity of presentation__ (Reviewers `s86g`, `Sj3P`), and the __quality of our results__ (all reviewers).

---

We include some additional experimental results (and the detailed settings) mentioned by Reviewer `s86g` in `response.pdf`:

- Figure 11 plots the valuation scores vs. error levels of distributions and fitted linear regression, for different number of vendors.
- Table 22 tabulates the "accuracy" (measured by Pearson coefficient) of our proposed valuation scores ($\hat{\Upsilon}$ in Equation 5) with mean and standard error over $10$ independent trials for up to $1000$ vendors.

---

### Decision · Program_Chairs · 2024-09-25

**Decision:**

Accept (poster)

**Comment:**

The paper studies the question of data valuation of a distribution when presented with a sample. The reviewers universally liked the paper and the contributions, I encourage the authors to take into account the points raised in the rebuttal and incorporate some of their new experiments into the final version of the paper.